# Boosting potassium-ion batteries by few-layered composite anodes prepared via solution-triggered one-step shear exfoliation

Yajie Liu [1], Zhixin Tai[1], Jian Zhang [2], Wei Kong Pang [1], Qing Zhang [1], Haifeng Feng [1], Konstantin Konstantinov [1], Zaiping Guo [1,3] & Hua Kun Liu [1]

Earth-abundant potassium is a promising alternative to lithium in rechargeable batteries, but a pivotal limitation of potassium-ion batteries is their relatively low capacity and poor cycling stability. Here, a high-performance potassium-ion battery is achieved by employing few-layered antimony sulfide/carbon sheet composite anode fabricated via one-step high-shear exfoliation in ethanol/water solvent. Antimony sulfide with few-layered structure minimizes the volume expansion during potassiation and shortens the ion transport pathways, thus enhancing the rate capability; while carbon sheets in the composite provide electrical conductivity and maintain the electrode cycling stability by trapping the inevitable by-product, elemental sulfur. Meanwhile, the effect of the exfoliation solvent on the fabrication of two-dimensional antimony sulfide/carbon is also investigated. It is found that water facilitates the exfoliation by lower diffusion barrier along the [010] direction of antimony sulfide, while ethanol in the solvent acts as the carbon source for in situ carbonization.

[1] Institute for Superconducting and Electronic Materials, Australian Institute for Innovative Materials, University of Wollongong, Innovation Campus, North Wollongong, NSW 2500, Australia. [2] College of Automotive and Mechanical Engineering, Changsha University of Science and Technology, Changsha 410114, China. [3] School of Mechanical, Materials, Mechatronic and Biomedical Engineering, Faculty of Engineering & Information Sciences, University of Wollongong, Wollongong, NSW 2522, Australia. These authors contributed equally: Yajie Liu, Zhixin Tai. Correspondence and requests for materials should be addressed to Z.G. (email: zguo@uow.edu.au)

Two-dimensional (2D) materials have various unique physical properties, which have prompted widespread successful application in the fields of catalysis, nanoelectronics, energy storage and conversion, etc. In particular, 2D materials present extensive prospects for application in energy storage and conversion due to their highly accessible surface area and fast charge transfer kinetics, so that they have been applied in a unique strategy to significantly enhance the rate performance of electrodes[1,2]. Potassium-ion batteries (KIBs) have attracted enormous attention due to their obvious advantages. Besides the abundance of potassium resources, the relatively lower redox potential of K/K$^+$ ($-2.93$ V vs. standard hydrogen electrode) than that of Na/Na$^+$ ($-2.71$ V), implies that KIBs could have a high-voltage plateau and high-energy density. Due to the large size of the K-ion, however, the insertion of K$^+$ into electrode materials is hindered, resulting in their relatively low capacity and poor cycling performance. The research on KIBs is still at an early stage, with the electrochemical reaction mechanism of most electrode materials unclear, and only a few cathode materials (such as Prussian blue[3], K$_x$MnFe(CN)$_6$[4], confined selenium (c-PAN-Se)[5]), and anode materials (graphite/carbon[6–9], Sn$_4$P$_3$/C[10,11]) could present reasonable capacity, although the cycling stability for all of them is far away from practical application. Therefore, further exploration of suitable electrode materials with high reversible capacity as well as good rate performance and excellent cycling stability is of great importance, and could possibly be achieved via the design and fabrication of 2D structured materials.

Among all the Sb-based anode materials, antimony trisulfide (Sb$_2$S$_3$) has drawn extensive attention[12,13], owning to its higher reversible theoretical capacity (946 mAh g$^{-1}$) compared to that of Sb anode (660 mAh g$^{-1}$) due to its theoretical accommodation of 12 moles of Li$^+$ or Na$^+$ per Sb$_2$S$_3$ mole. Better mechanical stability is also expected for Sb$_2$S$_3$ due to its smaller volume changes during charge/discharge than those of Sb anode. Moreover, the reversibility of sulfides is better than those of oxides (Sb$_2$O$_3$ and Sb$_2$O$_4$), resulting in relatively better cycling stability[14]. Improvement of the cycling stability and rate performance of Sb$_2$S$_3$ is necessary, however, to meet the requirements for real applications because of the unavoidable volume changes and limited ion/charge transfer. The bulk Sb$_2$S$_3$ crystal has a layered structure with zigzag sheets parallel to the $b$-axis, which makes the fabrication of 2D Sb$_2$S$_3$ possible. As far as we know, however, there has been no report on the electrochemical behavior of 2D Sb$_2$S$_3$ so far.

Exfoliation of layered bulk crystals to obtain monolayer or few-layer flakes has been a crucial technique in the fabrication of 2D materials[15], and it has been the primary technique in the synthesis of high-quality flakes for various applications. Among all the exfoliation techniques, high-shear mixing is a more effective approach than sonication for the large-scale fabrication of graphene[16] and other 2D materials (MoS$_2$ nanosheets[17], few atomic-layered LiCoO$_2$ material[2]), which could be a feasible and promising approach for industrial scale applications. Although this technique is promising, the challenges are still there, such as the relatively unclear mechanism of shear exfoliation and whether it is applicable to other layered structured materials.

In this study, the few-layered antimony sulfide/carbon sheet (SBS/C) anode is prepared via solution-triggered one-step high-shear exfoliation in order to boost the electrochemical performance of potassium-ion batteries (PIBs). The fatal issue of huge volume changes in Sb$_2$S$_3$ during electrochemical cycling could be solved by the design and fabrication of few-layered SBS/C anode, while the poor electrical conductivity of Sb$_2$S$_3$ can be improved by the incorporation of carbon via in situ carbonization in an ethanol-containing solvent. Moreover, the detected by-product S after cycling indicates that the irreversible conversion of SBS to S could be another reason for failure of SBS anode in KIBs, which could be overcome by the trapping effect of carbon sheets. In addition, the solvent effect on the exfoliation is also studied in order to optimize the structure and constitution of SBS/C composite. Based on density functional theory (DFT) calculations, the lower diffusion barrier of water than ethanol along the [010] direction of SBS crystal could give a pathway for facilitating exfoliation. Meanwhile, the ethanol in the solvent provides the carbon source for in situ carbonization. Ultimately, by employing a mixture of water (W) and ethanol (E) in a certain ratio as the exfoliation solvent, the as-prepared SBS/C (E/W = 2:1) composite anode delivers a specific capacity of 404 mA h g$^{-1}$ after 200 cycles (at a current density of 500 mA g$^{-1}$) and presents outstanding rate capability with 76% capacity retention at current densities from 50 to 500 mA g$^{-1}$.

## Results

**Electrochemical mechanism of SBS anode for K$^+$ storage and failure mechanism.** To understand the K$^+$ ion storage mechanism of Sb$_2$S$_3$ chemistry, the phase evolution of Sb$_2$S$_3$ for KIBs during discharge/charge was studied using in-operando synchrotron X-ray diffraction (XRD, $\lambda = 0.6888$ Å), ex situ XRD, and scanning transmission electron microscopy/selected area electron diffraction (SAED) (Fig. 1, Supplementary Fig. 1 in the Supporting Information, and Fig. 1c–e and Supplementary Fig. 2, respectively). Figure 1a presents the corresponding XRD patterns, which were collected at different stages in the first discharge cycle. In stage I (open circuit voltage (OCV) = 0.7 V), the peaks associated with the (301), (112), (400), (212), and (013) planes of Sb$_2$S$_3$ (PDF No. 040048897, Pnma (62)) shift toward lower detection angle, $2\theta$, indicating the insertion of K$^+$ into Sb$_2$S$_3$. Specifically, the 2D colorful contour plot using in situ diffraction patterns in Fig. 1b shows that the peak corresponding to (212) planes is shifted left and then becomes weakened in intensity. When the Sb$_2$S$_3$ anode is discharged from 0.7 to 0.5 V, the main peaks of Sb$_2$S$_3$ gradually become weakened, and two new peaks evolve at 15.17° and 15.68°, which are assigned to Sb (PDF No. 01-071-3736), and suggest the occurrence of a conversion reaction of Sb$_2$S$_3$. When further discharged to 0.1 V, two new peaks are generated at 13.4° and 13.7°, which can be ascribed to the formation of K$_2$S$_6$ (PDF No. 01-083-9589) as the intermediate state; two new peaks located at 14.17° and 14.46° correspond to the (131) and (221) planes of K$_2$S$_3$ (PDF No. 04-007-0574) as a final discharge product. At the same time, a new diffraction peak at 13.53° has developed, which can be indexed as the (220) peak of K$_3$Sb (PDF No. 01-078-6559). Due to the nanocrystallinity of the intermediate products of Sb and K$_2$S$_6$ (Supplementary Fig. 2), the corresponding diffraction peaks are weak and broad. In order to confirm the existence of intermediate products, ex situ SAED was conducted on the electrode after it was discharged to 0.5 V. As shown in the SAED pattern (Fig. 1a), the marked spots in orange are corresponding to the Sb (101) and (002) planes, which is consistent with the weak peaks (15.17° and 15.68°) in the synchrotron XRD pattern, and the diffused green rings with spots belong to the K$_2$S$_6$ (040), which corresponds to the peak at 13.4° in synchrotron XRD. Based on our in situ/ex situ XRD and SAED results, we propose that in KIBs, Sb$_2$S$_3$ may undergo K$^+$ intercalation reaction (1) followed by the conversion and alloying reactions (2, 3):

K$^+$ intercalation reaction:

$$\mathrm{Sb_2S_3} + x\mathrm{K}^+ + x\mathrm{e}^- \rightarrow \mathrm{K}_x\mathrm{Sb_2S_3} \quad (x{<}8) \tag{1}$$

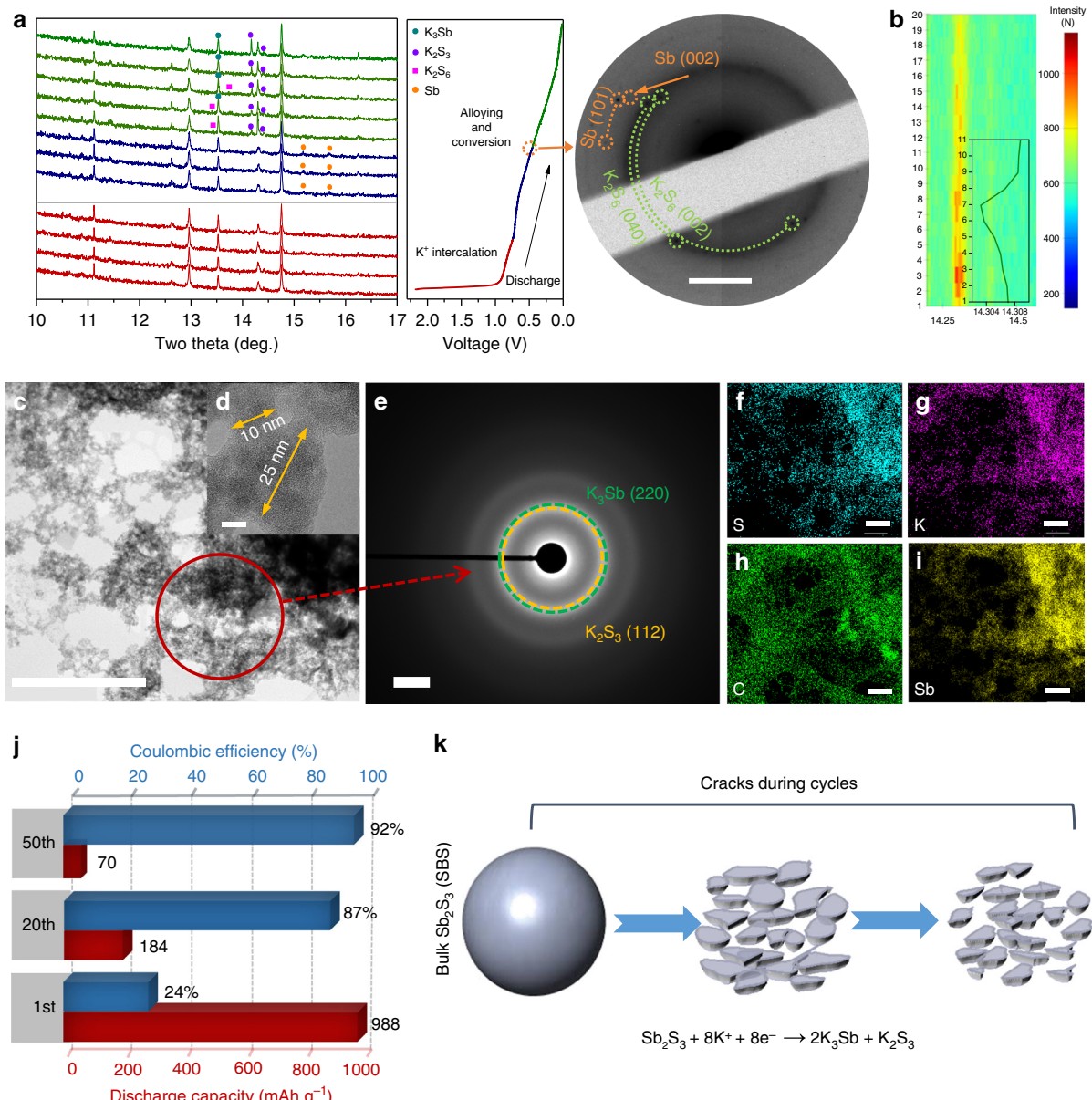

**Fig. 1** Investigation of the electrochemical mechanism of bulk $Sb_2S_3$ (SBS) anode and the failure mechanism. **a** In situ synchrotron XRD patterns of $Sb_2S_3$ electrodes upon K insertion at various potentials (left) and ex situ SAED pattern (right) (discharged to 0.5 V) with high-resolution image revealing weak reflections. **b** Image plots of the in situ XRD diffraction patterns of the (212) reflection of $Sb_2S_3$ during the intercalation stage and corresponding fitted peak (inset), indicating the peak shift. **c, d** TEM images of the first discharge product and high-resolution TEM image of the bulk $Sb_2S_3$ after potassiation. **e** SAED pattern of indicated area in (**c**). **f–i** STEM images with elemental mapping of sulfur, potassium, carbon, and antimony. **j** Discharge capacity and columbic efficiency of bulk $Sb_2S_3$ at different cycles. **k** Schematic illustration of pulverization of $Sb_2S_3$ during charge/discharge. Scale bars: 2 $nm^{-1}$ (**a**); 0.5 μm (**c**); 5 nm (**d**); 2 $nm^{-1}$ (**e**); 250 nm (**f–i**)

Conversion and alloying reactions:

$$K_xSb_2S_3 + 2xK^+ + 2xe^- \rightarrow 2Sb + 3K_xS \quad (x<2/3) \quad (2)$$

$$2Sb + 3K_xS + (8-3x)K^+ + (8-3x)e^- \rightarrow 2K_3Sb + K_2S_3 \quad (3)$$

Ex situ transmission electron microscopy (TEM) after discharge and charge were also employed to investigate the morphology and identify the phases of discharge/charge products (Fig. 1c–i and Supplementary Fig. 3). After conversion and alloying reactions of SBS with K ions, the discharge products show a connected globular morphology (Fig. 1c, d) with particle size around 20 nm instead of the initial microsized bulk material (Supplementary Fig. 4).

According to the electrochemical reactions above (Eqs. (1)–(3)), the volume expansion during discharging is about 300%, estimated based on the density differences between $Sb_2S_3$ (4.64 g $cm^{-3}$), $K_3Sb$ (2.24 g $cm^{-3}$), and $K_2S_3$ (2.12 g $cm^{-3}$). Meanwhile, the morphology of the $Sb_2S_3$ after charge also shows nanosized particles (Supplementary Fig. 3) with diameters around 25 nm (Supplementary Fig. 3d). The well-defined diffraction rings shown in the SAED pattern (Fig. 1e) undoubtedly reveal the polycrystalline nature of the charged product. Consistent with the XRD characterization (Fig. 1a), these rings can be satisfactorily indexed to (112) crystal planes of the $K_2S_3$ phase (PDF No. 04-007-0574) and (220) crystal planes of $K_3Sb$ phase (PDF No. 01-078-6559). Figure 1f–i and Supplementary Fig. 3e–h present energy dispersive X-ray spectroscopy elemental mapping images of discharged and charged products, respectively,

indicating uniform distributions of the S, K, and Sb in the discharged product, and homogenous distribution of Sb and S in the charged product. The electrochemical performance of bulk SBS is presented in Supplementary Fig. 5 and Fig. 1j. It presents a poor first cycle coulombic efficiency (around 23%) and poor electrochemical stability (from 988 down to 184 mA g$^{-1}$ after 20 cycles). The failure mechanism of commercial bulk SBS is primarily due to the huge pulverization during cycling, as illustrated in Fig. 1k. Continuous pulverization leads to, rupture of the solid-electrolyte interphase (SEI), decrease in electrical contact, and ultimately, fast deterioration in its electrochemical performance. In order to deal with the problem of serious pulverization, 2D material design was explored and found to be an effective way to overcome the problem, thus enhancing the performance of K-ion batteries.

**Few-layered SBS exfoliation and mechanism.** The orthorhombic crystal structure of Sb$_2$S$_3$ with *Pnma* phase (Fig. 2a), is composed of infinite chains of zigzag sheets of $(Sb_4S_6)_n$ along the *b*-axis. The weak bonding between the sheets makes the crystal cleavable along the *b*-axis direction[18,19]. The high-resolution TEM image of exfoliated Sb$_2$S$_3$ also confirms the possible cleavable direction, and Sb$_2$S$_3$ sheets/plates were obtained with (001) orientation (Fig. 2b). The size and thickness of exfoliated products were investigated via scanning electron microscopy (SEM), TEM, and atomic force microscopy (AFM). (Supplementary Fig. 6, Fig. 2b1, Supplementary Fig. 7, and Fig. 2c). From the comparison of SEM images (Supplementary Fig. 6), the average particle size of Sb$_2$S$_3$ obtained in ethanol is larger than that obtained in water. Figure 2c shows the thickness distribution of Sb$_2$S$_3$ nanosheets by counting more than 60 sheets for each sample collected from top solution. For the SBS exfoliated in water (W) and the mixed solution (E/W), the typical thickness of the nanosheets is mainly in the range of 2–8 nm and 4–15 nm, respectively, while for the SBS nanosheets obtained from ethanol solvent, the thickness is several times larger than that for the samples exfoliated in water or the mixed solvent, and is in the range of 26–55 nm. The typical AFM images and the cross-sectional height profiles conform the few-layered nature of the SBS nanosheet structures which were exfoliated in the mixed solution (E/W = 2:1) and pure water (W). From the XRD pattern of Sb$_2$S$_3$ after exfoliation in pure ethanol, the identified peaks are all consistent with of the standard PDF pattern, while for Sb$_2$S$_3$ exfoliated in water (W) or ethanol/water solution (E/W = 2:1), as shown in Fig. 2d, some of the characteristic peaks are relatively weaker or even not visible, indicating smaller crystal size, in good agreement with the thickness distribution shown in Fig. 2c. This difference in the peak intensity or peak disappearance in the samples could also be found in separate samples collected from same exfoliation solution (Supplementary Fig. 8a, Supplementary Fig. 9a). The AFM results together with XRD results indicated that the bulk Sb$_2$S$_3$ could be more easily exfoliated in water or water/ethanol solution than in pure ethanol. Water may play an important role in shear exfoliation. First-principles calculations were conducted in order to achieve further insight on the water and ethanol dynamics and to test that hypothesis. The calculated adsorption energies (Fig. 2e and Supplementary Note 1), suggest that water tends to be more easily adsorbed on the (010) surface of Sb$_2$S$_3$ than ethanol, which underpins further intercalation into the Sb$_2$S$_3$ crystal. Due to the unique open crystal structure on the (010) surface of SBS crystal, H$_2$O or CH$_3$CH$_2$OH could possibly intercalate and diffuse along the [010] direction of SBS. The diffusion barriers to water and ethanol along the [010] direction were also investigated by first-principles calculations as shown in Fig. 2f, Supplementary

Fig. 10 and Supplementary Note 1. The energy barriers for water and ethanol diffusion are 1.29 and 2.26 eV, respectively, indicating that the path along the [010] direction is more accessible for water than ethanol.

**Carbon sheet production during exfoliation.** Carbonization of ethanol could be occurring during high-shear exfoliation because the mechanical shearing and cavitation cause strong collisions between the active material and ethanol. Raman spectroscopy was performed to understand the composition of the exfoliated products (Fig. 3a, Supplementary Fig. 11) and to identify the carbon component in the samples exfoliated in ethanol-containing solvents. The peaks in the range of 200–500 cm$^{-1}$ correspond to the characteristic Raman shift of Sb$_2$S$_3$[20], while, the bands between 1300–1600 cm$^{-1}$ can be regarded as a D band (at around 1400 cm$^{-1}$) overlapping a G band (at around 1510 cm$^{-1}$), confirming the presence of amorphous carbon[21,22]. The intensity ratio of the D to the G band ($I_D/I_G$) is 1.4 for SBS/C exfoliated in ethanol and 1.6 for SBS/C exfoliated in the mixed solution. The higher intensity ratio for SBS/C (E/W = 2:1) indicates amorphous carbon structure in the composite with a higher content of plane defects or lattice edges. The Fourier transform infrared (FTIR) spectrum in Fig. 3b was collected to investigate the functional groups of the Sb$_2$S$_3$ exfoliated in different solutions. The transmission peaks at 690 and 1030 cm$^{-1}$ represent the symmetric bending of Sb–S and vibration of inorganic metal ions, respectively[23,24]. Compared with the commercial SBS and layered SBS exfoliated in water, the new peaks of the SBS/C exfoliated in ethanol-containing solvents in the range of 1500–1650 cm$^{-1}$ are attributed to the C=C vibration[25], and peaks in the range of 1000–1450 cm$^{-1}$ are attributed to CO (ester, ether, or hydroxyl) stretching and OH bending vibrations[26], respectively. The Sb 3d and C 1s XPS profiles of SBS/C and SBS were also obtained and deconvoluted to understand the composition and structure of the SBS/C composite. It was noted that the chemical state of Sb$_2$O$_3$ could be found for SBS exfoliated in water, with the peaks at 531 and 540.4 eV[27], indicating partial oxidation of Sb$_2$S$_3$ nanosheets, while for samples of SBS/C exfoliated in ethanol and ethanol/water solutions, these two peaks corresponding to Sb$_2$O$_3$ are absent (only 539.1 and 529.7 eV for Sb 3d$_{3/2}$ and Sb 3d$_{5/2}$ of Sb$_2$S$_3$)[13], suggesting that the oxide phase was not formed on the surface of Sb$_2$S$_3$ and that a relatively high-purity Sb$_2$S$_3$ phase can be produced when exfoliated in ethanol-containing solvent (Supplementary Fig. 12a–c). As for C 1s profile, peaks due to C–C (284.4 eV) and C=C (285.3 eV) are evident (Fig. 3c and Supplementary Fig. 12d)[28]; these bonds are mainly from the presence of carbon in the composite, with the content calculated to be about 90%. The intensities of the peaks of C=O and C–O (286.5 eV), and of COOR (288.4 eV)[29] are slightly higher for SBS/C exfoliated in ethanol/water solution than for its counterpart exfoliated in pure ethanol, and these functional groups evidenced in the C 1s spectrum are consistent with the FTIR results shown in Fig. 2b. These results (FTIR and C 1s XPS) suggest that there are carboxyl and hydroxyl groups on the surface of the amorphous carbon that can be ascribed to the incomplete carbonization at a relatively low exfoliation temperature. From the TEM and energy dispersive spectroscopy (EDS) results (Fig. 3d, e and Supplementary Fig. 13), it is found that the carbon sheets are relatively thinner compared to the SBS sheets. In addition, from visual inspection of the exfoliated solutions with different solvents, the exfoliated solutions with ethanol are apparently darker than that in water (Supplementary Fig. 14), due to the presence of carbon.

**Electrochemical properties of few-layered SBS and SBS/C electrodes.** Electrochemical properties of the layer-structured SBS

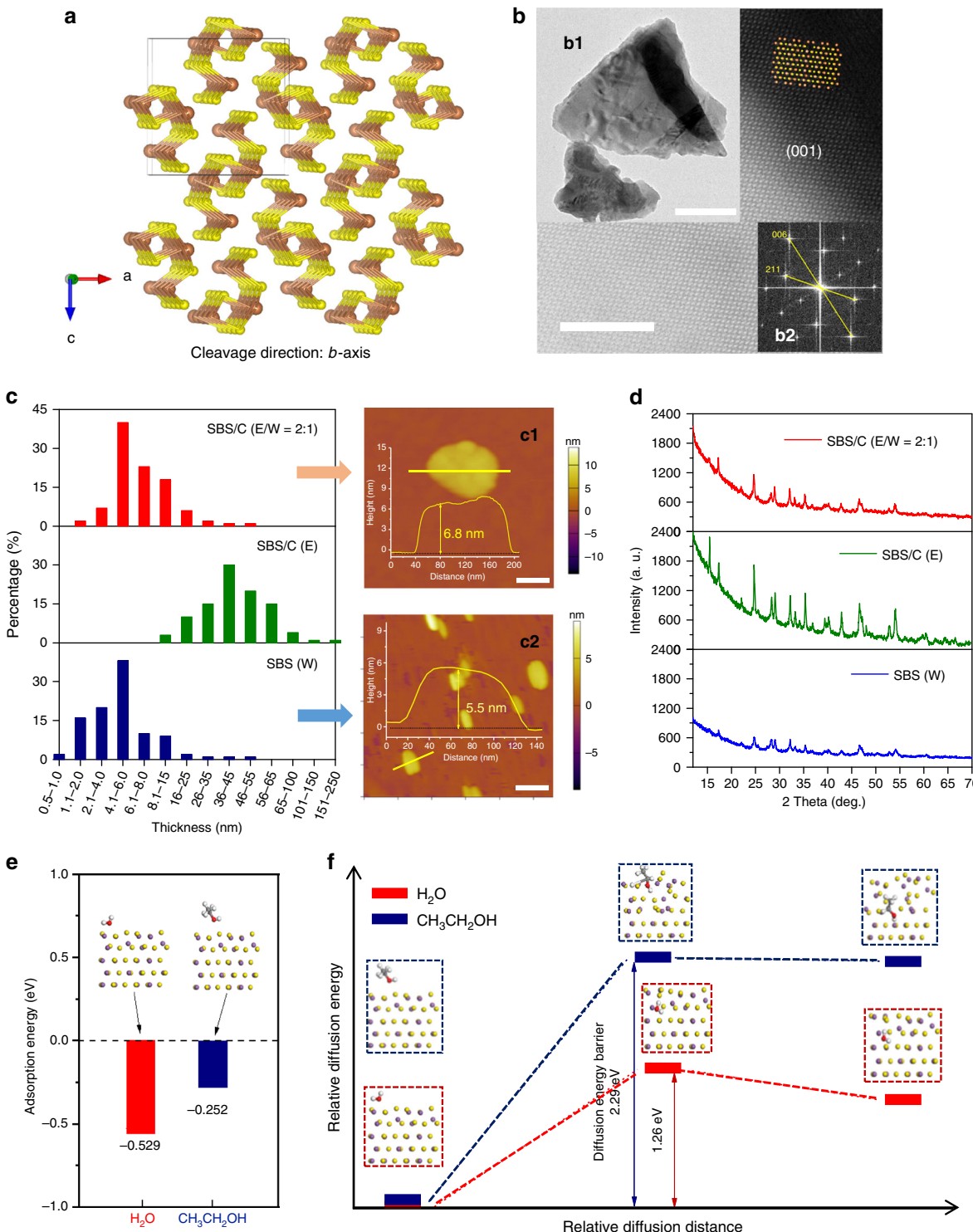

**Fig. 2** The characterization of exfoliated SBS and the solvent effect for exfoliation. **a** The crystal structure of bulk $Sb_2S_3$ with cleavage tendency along b-axis. **b** High-resolution TEM image of exfoliated $Sb_2S_3$ with inset TEM image (**b1**) and FFT pattern (**b2**). **c** Histograms of the thickness distribution of SBS nanosheets prepared with different solvents (mixed ethanol and water (E/W = 2:1), pure ethanol (E), and pure water (W)), and typical AFM images with inset cross-sectional height profiles of SBS nanosheets obtained from SBS/C (E/W = 2:1, c1) and SBS (W, c2). **d** XRD patterns of exfoliated $Sb_2S_3$ with different exfoliation solvents (mixed ethanol and water (E/W = 2:1), pure ethanol (E), and pure water (W)). **e** Adsorption energy of water and ethanol molecules on $Sb_2S_3$ (010) surface. **f** Relative diffusion energy barrier of water and ethanol molecules diffusing along the [010] direction from the $Sb_2S_3$ (010) surface to a position between the second and the third layers of $Sb_2S_3$ (110) slab, calculated with DFT and presenting the diffusion transition state, initial state, and final state with the indicated crystal structures. Scale bars: 5 nm (**b**); 200 nm (**b1**); 100 nm (**c1**); 200 nm (**c2**)

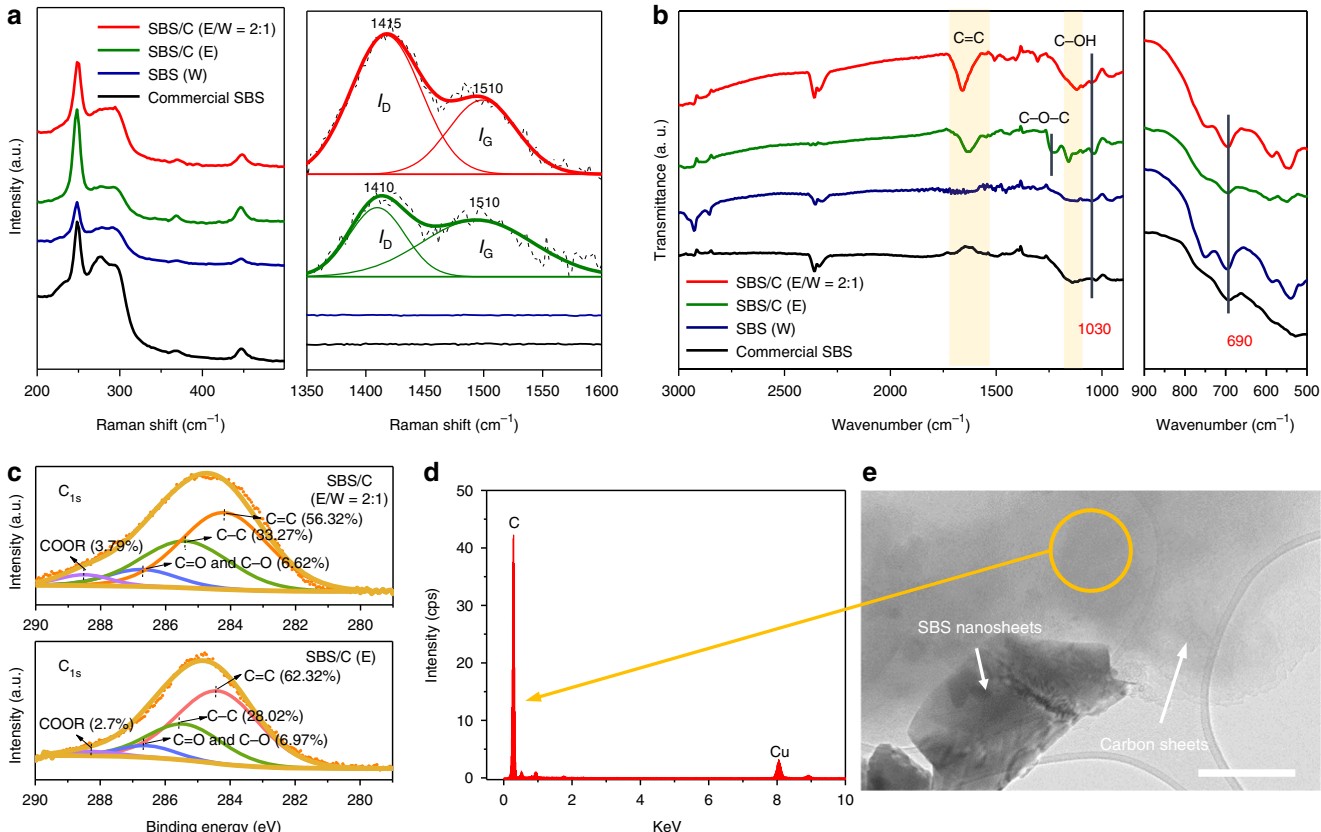

**Fig. 3** Characterization of carbon sheets. **a** Raman spectra of exfoliated products in different solvents, SBS (W), SBS/C (E), and SBS/C (E/W = 2:1) compared with commercial SBS, indicating the generation of amorphous carbon during exfoliation in ethanol-containing solvents. **b** FTIR spectra of commercial SBS, SBS (W), SBS/C (E) and SBS/C (E/W = 2:1). **c** XPS analysis of C 1s peaks of SBS/C (E/W = 2:1) and SBS/C (E). **d** EDS spectrum of the indicated region in (**e**), indicating the carbon sheets. **e** TEM image of exfoliated SBS/C composite (E/W = 2:1). Scale bar: 200 nm (**e**)

and SBS/C anodes were investigated in the range of 0.01–2 V (vs. $K^+/K$). The capacities of SBS/C electrodes were calculated on the basis of the total mass of SBS and carbon. Supplementary Fig. 15 presents typical cyclic voltammograms of the few-layered SBS and bulk SBS for the initial three cycles at 0.1 mV s$^{-1}$. In the first cycle of few-layered SBS (NS-3-W), three cathodic peaks at 0.78, 0.45, and 0.31 V are attributed to the intercalation process and the formation of SEI, the conversion reaction with sulfur in SBS, and the alloying of K with Sb, respectively, which are assigned based on in situ XRD analysis and research on SBS anode in lithium-ion batteries (LIBs) and sodium-ion batteries[12,20,30]. For the following cycle, all of the peaks are almost overlap for the few-layered SBS, suggesting good reversibility. In order to understand how the exfoliation solvents influence the electrochemical performance of the SBS samples, the cycling stability and rate capability of the different SBS electrodes are compared (Fig. 4a–d). All samples collected from ethanol in different sections (NS-1-E, NS-2-E, and NS-3-E) present better cycling stability than the samples collected from water (NS-1-W, NS-2-W, and NS-3-W), mainly due to the contribution of carbon from the carbonization of ethanol. The charge/discharge curves of SBS/C (NS-2-E) and bulk SBS at different cycles are compared in Fig. 4e and Supplementary Fig. 16. The results show that there is a new voltage plateau (around 1.7 V) after a certain number of cycles for SBS/C accompanied by slightly increased capacity, while for bulk SBS, there is no new voltage plateau and the capacity drops quickly. To explore the chemical state of SBS, we analysed the S 2p peaks of the SBS/C (NS-2-E) and bulk SBS electrode surfaces via ex situ XPS (Fig. 4f, g). The two peaks of SBS/C and bulk SBS located at 161.55 and 162.37 eV show the presence of $S^{2-}$[13], while the two peaks at 169

and 170.5 eV are likely to reflect the $-SO_2-$ fragments due to the decomposition of the bis(fluorosulfonyl)imide (FSI-) anion in the SEI film[31]. The high ratio/content of $-SO_2-$ on the surface of bulk SBS after the 50th charge indicates a thicker SEI film compared with the layered SBS/C electrode due to the continuous pulverization of bulk SBS during cycling. In addition, it should be noted that the element sulfur was detected in both electrodes, with its peak located at around 164 and 165 eV[32], and the ratio of elemental $S^0$ to $S^{2-}$ on the electrode surface after the 50th charge are 16.5% and 9.04% for SBS/C (E) and SBS (bulk), respectively (Supplementary Table 1 and Supplementary Note 2). The high intensity of elemental sulfur in SBS/C could be ascribed to the carbon trapping effect, in which the carbon sheets produced in SBS/C composite may play a crucial role by trapping the polysulfides and preventing them from dissolving in electrolyte and creating the shuttle effect during discharge/charge. Therefore, in the SBS/C composite, the carbon sheets in composite not only increase the electrical conductivity, but also prevent the loss of active material ($Sb_2S_3$ or S). On the other hand, without the protection of carbon sheets in bulk SBS and layered SBS (W), the by-product sulfur that is produced could cause the shuttle effect and capacity decay. In terms of the rate capability of SBS/C and layered SBS, although SBS/C shows better rate performance than layered SBS in the low current density range (from 20 to 300 mA g$^{-1}$), layered SBS presents slightly better rate capability at high current densities (from 500 to 1000 mA g$^{-1}$). Carbon sheets in the SBS/C composite play a dominant role in improving the electrical conductivity, which enhances the rate performance at low current densities. At high current densities, however, the much thicker SBS sheets in the composite (around 45 nm in

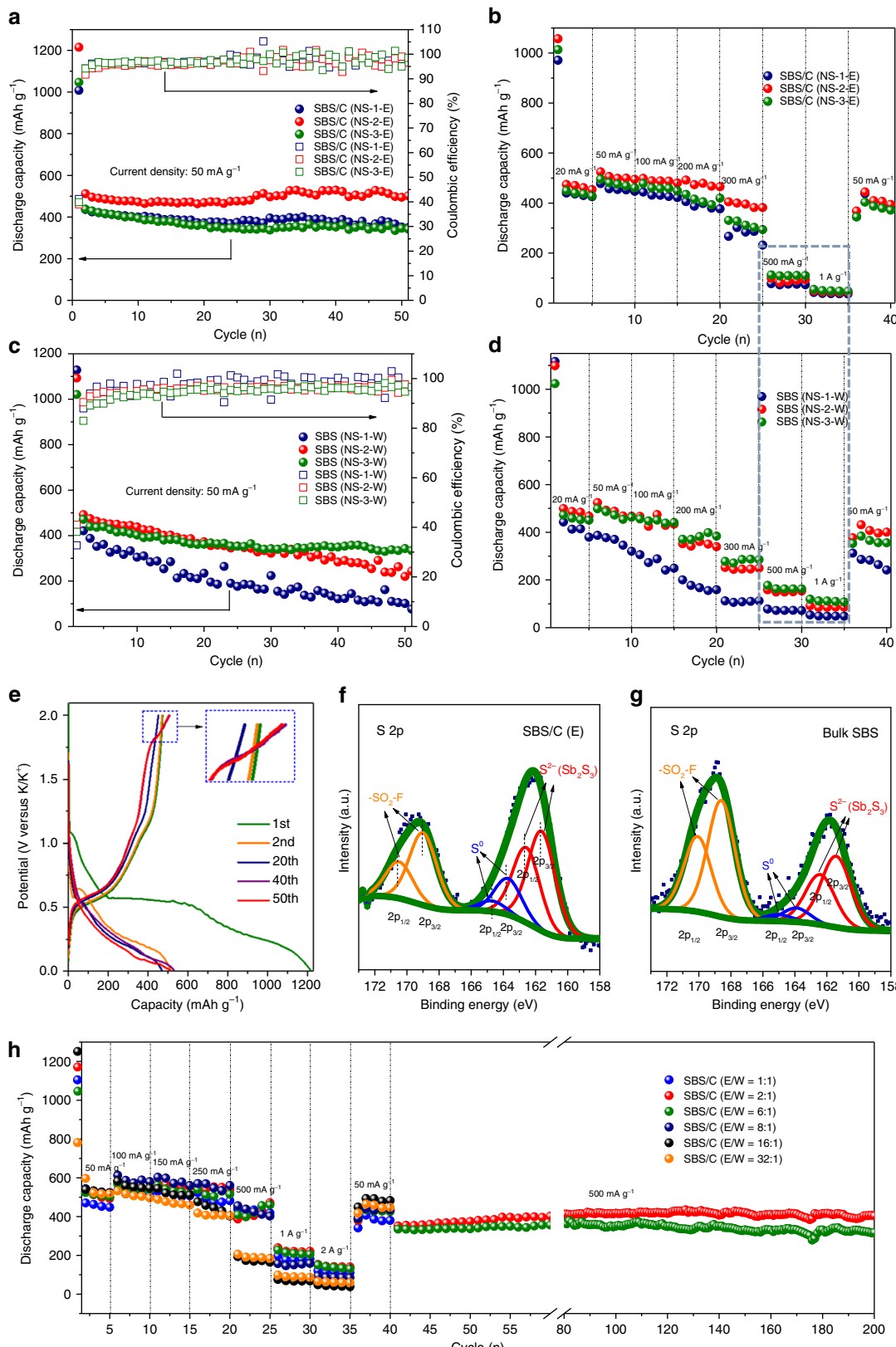

ethanol vs. around 5 nm thickness in water) become the major limitation for ion diffusion. With the decreased thickness of the sheets in layered SBS, the shortened ion diffusion pathways lead to a much improved K-ion diffusion coefficient (Supplementary

Fig. 17 and Supplementary Note 3), resulting in improved rate capability, especially at high current densities. (Fig. 4d).

As it was concluded above that water could facilitate the exfoliation by easily absorbing and diffusing into SBS crystal

(Fig. 2e, f) and ethanol works as a carbon source, it is reasonable to expect that the mixed solvent of ethanol/water would be the best choice, which not only maintains the few-layered structure of SBS but also introduces carbon sheets at the same time. Therefore, in order to optimize SBS anode for achieving outstanding electrochemical performance, layered SBS/C samples were synthesized via exfoliation in ethanol/water mixtures with various ratios (E/W = 1:1, E/W = 2:1, E/W = 6:1, E/W = 8:1, E/W = 16:1, E/W = 32:1). In Fig. 4h, from the cycling response at different current densities (from 50 mA g$^{-1}$ to 2 A g$^{-1}$), we find that the SBS/C (E/W = 2:1) and SBS/C (E/W = 6:1) electrodes present the best rate capability. Moreover, it was found that the initial discharge capacity of SBS/C electrodes at 50 mA g$^{-1}$ is slightly lower than the following capacity at 100 mA g$^{-1}$ (Fig. 4h) in the rate test, and a similar unusual phenomena can also be observed in Fig. 4b, d, which can be attributed to the activation process of SBS and SBS/C electrodes in the initial cycles and could be avoided by extending the standing time of fresh cells (Supplementary Fig. 18).

The carbon content in each composite has been investigated in Supplementary Fig. 19 and Supplementary Note 4, (with 3.82% in SBS/C (E:W = 1:1), 3.93% in SBS/C (E:W = 2:1), 3.62% in SBS/C (E:W = 6:1), 3.72% in SBS/C (E:W = 8:1), 4.17% in SBS/C (E:W = 16:1), and 4.94% in SBS/C (E:W = 32:1)), indicating that there is not much difference in carbon content among these samples. In order to distinguish the influence of carbon content and the thickness of SBS sheets on the electrochemical performance of composite electrodes, we designed and fabricated SBS/C composites via a two-step shear exfoliation. It was found that the cycling stability and rate performance improved with decreasing thickness of the SBS sheets, and among them the SBS 6000 electrode shows the best electrochemical performance. We then investigated the influence of the carbon content based on the same thickness of SBS. Supplementary Fig. 20 suggests that although the presence of carbon sheets improved the cycling stability and rate capability significantly, the carbon content in the composites (in the range from 3.5 to 5 wt%) plays a negligible role in influencing the electrochemical performance. Therefore, the differences in electrochemical performance of the few-layered SBS/C electrodes shown in Fig. 4h are mainly caused by the thickness of SBS. Supplementary Figs. 21, 22 show the variations in the size and thickness distributions of the exfoliated sheets in these SBS/C electrodes via SEM and AFM. Less thickness of layered SBS in a composite will lead to the short diffusion time according to the equation $t = L^2/D$[33] (Supplementary Fig. 23) (where $t$ is the diffusion time, $L$ is the diffusion length or the thickness of SBS, and $D$ is the K-ion diffusion constant in SBS), which results in the simultaneous transfer of K ions with improved rate performance. The cycling performances of SBS/C (E/W = 2:1) and SBS/C (E/W = 6:1) electrodes were compared after the rate test, and better cycling stability of SBS/C (E/W = 2:1) was achieved, with a high reversible capacity of 404 mAh g$^{-1}$ after 200 cycles (Fig. 4h). The long-term cycling performance of SBS/C (E/W = 2:1) electrode was further investigated (Supplementary Fig. 24). It shows excellent cycling stability and high-capacity retention of 79% after 1000 cycles (at a current density of 1 A g$^{-1}$). Meanwhile, we also compared the SBS/C (E/W = 2:1)

electrode with previously reported state-of-the-art anodes[11,34–42] for KIBs, excluding carbon-based anodes (Fig. 5). It is shown that our few-layered SBS/C electrode could deliver the highest reversible capacity with unrivaled cycling stability among all the anode materials so far (excluding carbon/graphite anode). The superior cycling stability and rate capability of SBS/C (E/W = 2:1) are mainly due to the synergetic effects between few-layered structured SBS and the carbon sheets in the composite, which not only promote ion/electron transfer, but also maintain the electrode/structure stability and electrode reversibility. Here, we exclude the carbon/graphite anodes from the comparison of electrochemical performance because of their limitations as anode for PIBs. The very low theoretical capacity (279 mA h g$^{-1}$, 30% less than that of LIB) and poor capacity retention of the graphite anode mean that it cannot rival nongraphite anode in KIBs. Although amorphous carbon electrode present higher reversible capacity than graphite (250 vs. 200 mAh g$^{-1}$) with relatively better cycling retention, the electrochemical behavior is more like capacitor behavior, with a sloped, inconspicuous, and relatively high voltage plateau.

## Discussion

According to the research presented above, high-performance PIBs with a composite of few-layered antimony sulfide/carbon sheets (SBS/C) as anode are introduced. The SBS/C composite was fabricated via one-step high-shear exfoliation in an ethanol/water solvent (ratio E/W = 2:1). Compared with commercial bulk SBS, the few-layered SBS/C could effectively deal with the issues related to the huge volume changes of $Sb_2S_3$ during charge/discharge and its poor electrical conductivity. Few-layer structured SBS anode could minimize the absolute volume changes of SBS and facilitate ultrashort ion transport paths compared with the bulk material. The microsized carbon sheets in the composite not only provide electrical conductivity, but also avoid the loss of active material by trapping the element S that is inevitably produced due to the irreversible reaction between $K_2S_3$ and $Sb_2S_3$. After investigating the solvent effect (water and ethanol) on exfoliation, it was found that water could facilitate the exfoliation to produce few-layered SBS based on the experimental results and DFT calculations, while ethanol could promote carbon sheet generation during exfoliation due to the carbonization. The electrochemical performance of layered SBS/C anode was further optimized by investigating the solvent ratio of ethanol/water for exfoliation. With the cooperative action of water and ethanol in a certain ratio as exfoliation solvent, the obtained SBS/C (E/W = 2:1) composite anode could deliver a reversible capacity as high as 404 mA h g$^{-1}$ after 200 cycles (at a current density of 500 mA g$^{-1}$) and present excellent rate performance with 76% capacity retention from current densities of 50–500 mA g$^{-1}$.

At the same time, in order to obtain insight on the electrochemical behavior of $Sb_2S_3$ in PIBs, the electrochemical reaction mechanism of the $Sb_2S_3$ electrode during charge/discharge was investigated by in situ XRD, ex situ XRD, and TEM. The two-stage reactions of SBS in the PIB are proposed. Unlike the reactions of SBS in LIBs/SIBs, it clearly presents an intercalation step before the conversion/alloying reactions. According to the final

**Fig. 4** Electrochemical properties of few-layered SBS and SBS/C electrodes. Comparison of (**a**) cycling performance and (**b**) rate performance of SBS/C (NS-1-E), SBS/C (NS-2-E), and SBS/C (NS-3-E) electrodes, which were exfoliated and collected in ethanol. Comparison of (**c**) cycling performance and (**d**) rate performance of layered SBS (NS-1-W), SBS (NS-2-W), and SBS (NS-3-W) electrodes, which were exfoliated and collected in water. **e** Discharge/charge curves at different cycles of SBS/C (NS-2-E) composite at 50 mA g$^{-1}$. Ex situ X-ray photoelectron spectroscopy (XPS) of the S 2p peaks of (**f**) SBS/C (NS-2-E) and (**g**) bulk SBS electrodes after the 50$^{th}$ charge. **h** Rate capabilities of SBS/C electrodes exfoliated with different solvents (E/W = 1:1, E/W = 2:1, E/W = 6:1, E/W = 8:1, E/W = 16:1, E/W = 32:1) obtained at various charge and discharge current densities (at 50, 150, 300, 500, 1000, and 2000 mA g$^{-1}$) and their cycling performance after rate testing at a current density of 500 mA g$^{-1}$

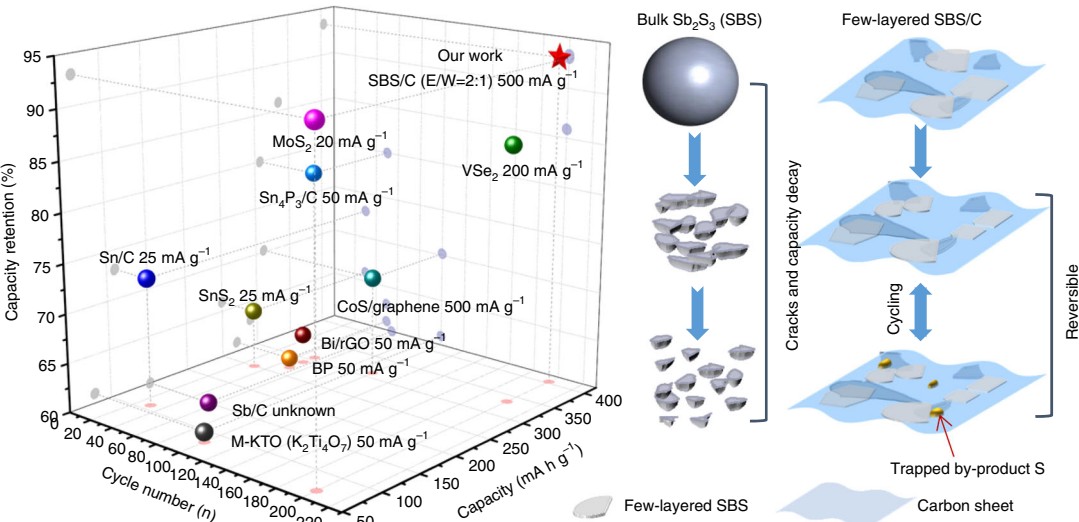

**Fig. 5** State-of-the-art reported anodes (except for carbon/graphite) for KIBs[11, 34–42], and schematic illustration of working mechanism of bulk SBS and few-layered SBS/C

discharge products, the calculated theoretical capacity of SBS in KIBs is as high as 630 mA h g$^{-1}$.

Moreover, in terms of fabrication, the solvent effect was discussed for fabricating different thicknesses of layered SBS via shear exfoliation, which gives guidance for exfoliating other materials with control of thickness. The carbonization of organic solvent during high-energy shear exfoliation was proposed and demonstrated for the first time in our work. This strategy for the fabrication of few-layered material/carbon composite can be extended to other layered crystals, and the shear exfoliation may open a new path towards carbon composite fabrication with selected organic solvents.

## Methods

**Materials**. All the involved chemicals were analytic grade and they were purchased from Sigma Aldrich. They were all directly used without any further purification.

**Shear exfoliation**. SBS and SBS/C were synthesized via a modified shear exfoliation method using the commercial bulk materials[2]. The L5M high-shear laboratory mixer that was employed is made by Silverson Machines Ltd., UK. Typically, the screw-on slotted and interchangeable disintegrating head equips a rotor with 30 mm in diameter. The gap between the head screen and rotor is about 0.05 mm (illustrated in Supplementary Fig. 25). When it is in operation, the high rotor speed ($N$) creates a high-shear rate ($\gamma$) within the gap. More specifically, in synthesis, the bulk SBS (5 g) was put into a beaker (250 ml) containing ethanol, water, or a mixed solution of ethanol and water (1:1, 2:1, 6:1, 8:1, 16:1, and 32:1) (total volume of 200 ml). The mixtures were kept at room temperature for 2 h before exfoliation. Then, the mixer head was lowered, immersed in the solution, and then rotated at 6000 rpm for a continuous 30 min. During mixing, the beaker was fixed in a water bath with the initial temperature of 20 °C. After 24 h of standing after mixing, the obtained dispersion was divided into three samples (top dispersion: NS-3; middle dispersion: NS-2; and bottom dispersion: NS-1). For the SBS exfoliated in ethanol, the three separated samples were denoted as NS-1-E, NS-2-E, and NS-3-E, while for the SBS exfoliated in water, the samples were denoted as NS-1-W, NS-2-W, and NS-3-W. These samples were then collected via filtration. In the case of the sample collection from the mixed exfoliation solution, the procedure was similar, but collection was only done from the top of the dispersion for filtration. For the two-step exfoliation, different thicknesses of SBS in water were fabricated with adjusting the rotation rate (4000, 5000, 6000, and 7000 rpm) (SBS 4000, SBS 5000, SBS 6000, and SBS 7000). Then, the solution product of SBS 6000 was chosen as target sample, and different amounts of ethanol (ethanol/water ratio: 1:1, 4:1, and 16:1) were added for step-two exfoliation.

**Materials characterization**. The microstructure/morphology of the as-prepared Sb$_2$S$_3$ bulks and nanosheets was investigated by XRD (GBC MMA) with Cu K$\alpha$ radiation; field-emission SEM (FESEM) (JEOL 7500); TEM (JEOL ARM-200F) with high-resolution TEM (HRTEM), and Raman spectroscopy (Jobin Yvon HR800) employing a 10 mW helium/neon laser at 632.8 nm. A commercial AFM (Asylum Research MFP-3D) was used to measure the morphology and thickness of the SBS nanosheets in trapping mode. An Al coated $n$-silicon probe with resonance frequency of 204–497 kHz and force constant of 10–130 N m$^{-1}$ was used in the AFM measurements. For synchrotron X-ray powder diffraction, a specially modified CR2032 coin cell was used with holes on both sides. In situ synchrotron XRD measurements were then performed at the Powder Diffraction beamline (Australian Synchrotron), and the XRD patterns were conducted at 0.688273 Å (determined using LaB$_6$, NIST SRM 660b).

**Electrochemical measurements**. The commercial SBS, exfoliated SBS and SBS/C electrodes were assembled in a glove box into coin cells (CR2032). For the anode preparation, slurry containing 60 wt% active material, 20 wt% Super P, and 20 wt% carboxymethyl cellulose was dissolved in an aqueous solution. Then, the working electrodes were prepared by coating the slurry on a copper foil current collector and drying it at 70 °C for 12 h. The loading mass of the active materials (SBS or SBS/C) was around 1 mg cm$^{-2}$. 1 M potassium fluorosilicate (KSiF$_6$) in ethylene carbonate/ propylene carbonate (1:1 V/V) was applied as the electrolyte. The as-fabricated coin cells were charged and discharged in the voltage range of 0.01–2 V for the SBS anode using a Neware instrument.

**Theoretical calculations**. The calculations were performed based on the DFT approach[43] using the DMol$^3$ package. The exchange-correlation interaction was tested by using the generalized gradient approximation (GGA) with the Perdew-Wang 91 (PW91) function[44]. The double numerical basis with polarized orbital (DNP) was specified as the atomic orbital basis set[45]. The Brillouin-zone was sampled using the Monkhorst-Pack grid of special $k$-points[46]. The convergence tolerances of energy, the maximum force, and the maximum displacement were $1.0 \times 10^{-5}$ Ha, 0.002 Ha Å$^{-1}$, and 0.005 Å for, respectively.

## Data availability

The data that support the findings of this study are available from the corresponding author on reasonable request.

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

## Acknowledgments

This research has been conducted with the support of an Australian Government Research Training Program Scholarship (Y.L.). Support from the Australian Research Council (ARC) through a Future Fellowship project (FT150100109), (FT160100251), and DP170102406 and Auto CRC Project 1-117 are gratefully acknowledged. The authors would also like to thank the Electron Microscopy Centre (EMC) at the University of Wollongong for the electron microscopy characterizations, Prof. Dianwu Zhou in Hunan University for the support of the Dmol3 software, Dr. Justin Kimpton of Australian Synchrotron support on synchrotron XRD measurements and Dr. Tania Silver for critical reading of the manuscript and valuable remarks.

## Author contributions

Y.L. and Z.T. proposed, conceived the ideas; Y.L. designed and performance the experiment and do most of characterizations; Y.L. and Z.T. analysed all the data; Y.L. drafted the manuscript. Besides, J.Z. performed the DFT calculations; W.P., and Q.Z. performed the in situ XRD test and W.P. checked and revised the description of the in situ data; H.F. performed the AFM characterization. K.K participated in discussions and provided the comments. Z.G (corresponding author) supervised the study, performed manuscript editing and provided comments and suggestions. H.L. gave the support and provided the comments and suggestions.

## Additional information

**Competing interests:** The authors declare no competing interests.

