## [Peer Review File · Nature Communications]

Reviewer #1 (Remarks to the Author):

This manuscript reported the high-performance few-layered $\text{Sb}_2\text{S}_3/\text{C}$ composite anode for potassium ion battery, which is fabricated via novel technique of one-step shear exfoliation. The well-developed few-layered structure of SBS anode enhanced the potassium ion battery performance, with capacity superior to the traditional graphite anode for LIBs. Based on the abundance and the low cost of potassium than lithium, this system could be a promising alternative for LIBs. Besides, the achievement and new findings regarding the in-situ carbonization during exfoliation and the small amount of S production during electrochemical cycling in potassium ion battery are all impressive and interesting, which could be beneficial for structure design and deep mechanism understanding. Overall, this manuscript is well organized and written. And the experimental aspects of the research are thoroughly detailed and well explained. I recommend its publication after addressing the following comments and I am sure the following revisions would serve to further strengthen the manuscript.

1. Please label the figures in sequence according to the description in context. For example, description of Figure S2i in advance of Figure S2d-g is not appropriate.
2. In Figure 2(e), there is no X axis title; it is a bit difficult to understand. Please reorganize this figure.
3. In Figure 3a, please enlarge the note of ID and IG for clear presentation.
4. The authors mentioned that the shear exfoliation is efficient than sonication. And the A2B3 crystals such as Sb_2S_3 could be exfoliated via high shear exfoliation. However, from context and the supporting information, barely no information focuses on the working mechanism of shear exfoliation and the devices they used. Please provide more description/discussion for better understanding.
5. In addition, it should be specified whether the discharge capacity is calculated based on the total mass (Sb_2S_3 and carbon) or the weight of active material Sb_2S_3 .
6. In Figure 4, the first discharge capacity for the SBS and SBS/C electrode is very high, nearly 1200 mAh g^{-1} , which is higher than the theoretical capacity. Please give explanation. And why is the first coulombic efficiency so low?
7. Although few-layered SBS/C anode shows good performance in K ion batteries, this manuscript does not provide some important parameters of the electrodes, such as loading amount of active materials. Without these parameters, it is not fair to compare the performance of the electrode with those of the other electrodes.
8. In Figure 4 (h), the SBS/C (16:1 and 32:1) electrodes exhibit a slightly higher capacity at 50 mA/g after rate test, compared with other electrodes. The authors are encouraged to discuss possible reasons for this phenomenon.

Reviewer #2 (Remarks to the Author):

This paper reports that the few-layered SBS/C composite was prepared by one-step high-shear exfoliation in ethanol/water solvent. The authors used the few-layered SBS/C composite as anode of potassium ion battery, achieving outstanding performance with high reversible capacity, good rate capability, long cycling stability. The work is interesting and this manuscript is logically-organized well. However, this manuscript needs major revisions before acceptance, due to the following reasons:

- (1) The authors claim that the obtained materials are few-layered $\text{Sb}_2\text{S}_3/\text{C}$. However, from all TEM and SEM, it is not obvious. Please provide more clear evidence.
- (2) The first charge-discharge curve is very important to explain its electrochemical properties. Please give the first charge-discharge data in Fig. 4 e.
- (3) The carbon in the composite can greatly affect the electrochemical properties of the battery, the authors should make sure a similar mass content of Carbon in these SBS/C composites as far as possible. Otherwise, this comparison makes no sense.

- (4) Whether the good electrochemical performance was caused by different carbon content or because of the different layers, the reason was unknown. The author should make more in-depth comparison and discussion.
- (5) In figure 4b and d, discharge capacity of the ordinate is set to -200mAhg^{-1} , which is suspected to be intentionally misleading readers. Please correct it.
- (6) The quality of Figure 2c should be improved. Its axis is blurred, please adjust it.
- (7) As we known, the discharge capacity at lower current density often is higher than that of high current density. However, Fig. 4,b,d, h shows an opposite result. The authors should further explain the unusual phenomena.
- (8) If possible, the long cycle performances should be further investigated by the authors to show the better stability of SBS/C.

Reviewer #3 (Remarks to the Author):

The manuscript of Y. Liu et al. describes a facile synthesis of a few layered antimony sulphide/carbon sheet (SBS/C) composite and its electrochemical performance as anode for potassium ion batteries. Unlike sodium ion batteries that have attracted significant attention from the research community in the last five years, potassium-ion cells are in an earlier stage of evolution, although the interest in these systems is clearly increasing. A authors describe the novel and interesting results on the preparation of the anode material via simple one-step high-shear exfoliation in ethanol/water solvent and its superior cycling stability, rate capability and impressive electrochemical capacity (above 400mAh/g after 200 cycles at current density of 500mA/g). All described characteristics of the material make it attractive and promising as an active component of the anode in potassium ion batteries. Results presented by Y. Liu et al. are reliable, fully consistent and highly interesting in many relations, including the studies of mechanism of the exfoliation and potassium interaction with anode material during the cycling. Presented results and conclusions will be of interest to others in the community and the wider field.

However, some points have to be clarified and additional data should be provided before I recommend this manuscript for publication in Nature Communications.

1. Authors suggest the formation of the Sb and K_2S_6 phases during the potassium intercalation to the anode based on the XRD data presented on the Figure 1a. However, the corresponding peaks have a very low intensity, and it seems impossible to distinguish them from the noise of the diffractogram. This point needs to be clarified and discussed, and probably additional frame with a higher resolution should be presented in Figure 1.
2. In order to gain insight on the chemical composition and structure of the carbon in SBS/C composite the additional XPS studies (carbon 1s peak) should be provided. Additionally, the probable oxidation (or reduction) of the antimony during the high-shear exfoliation in ethanol/water solvent and formation of the antimony (III or V) oxides (or elemental antimony) on the surface of the Sb_2S_3 nanosheets should be discussed based on the XPS (Sb 3d peak) studies.
3. The cyclic voltammograms were assigned based on in-situ XRD analysis and batteries research on SBS anode in lithium and sodium ion batteries (page 16). However, the presented references 12 and 20 correspond to the Sb_2S_3 studies as sodium ion batteries anodes. I would recommend to cite additionally the article of P.V. Prikhodchenko et al. (Chemistry of Materials, 2012, 24, pp 4750–4757) describing the $\text{Sb}_2\text{S}_3@\text{Graphene}$ for a superior lithium battery anode.
4. The elemental sulfur and SO_2 content in the SBS/C composite are discussed on the page 17 but all values are presented only in ESI. Some conclusions like " it should be noted that the element sulfur was detected in both electrodes and it shows much higher intensity/content in SBS/C than bulk SBS" should be supported with values of the sulfur content directly in the manuscript.
5. The similar comment could be made regarding the discussion on the page 19 on the carbon content in the SBS/C composites. Values of the carbon content should be introduced into the text of the manuscript.

6. I would suggest to revise the structure of the manuscript. The discussion part of the manuscript is in fact conclusions, but the chapter "results" actually contains "results and discussion".

Reviewers' comments:

Reviewer #1 (Remarks to the Author):

This manuscript reported the high-performance few-layered Sb₂S₃/C composite anode for potassium ion battery, which is fabricated via novel technique of one-step shear exfoliation. The well-developed few-layered structure of SBS anode enhanced the potassium ion battery performance, with capacity superior to the traditional graphite anode for LIBs. Based on the abundance and the low cost of potassium than lithium, this system could be a promising alternative for LIBs. Besides, the achievement and new findings regarding the in-situ carbonization during exfoliation and the small amount of S production during electrochemical cycling in potassium ion battery are all impressive and interesting, which could be beneficial for structure design and deep mechanism understanding. Overall, this manuscript is well organized and written. And the experimental aspects of the research are thoroughly detailed and well explained. I recommend its publication after addressing the following comments and I am sure the following revisions would serve to further strengthen the manuscript.

1. Please label the figures in sequence according to the description in context. For example, description of Figure S2i in advance of Figure S2d-g is not appropriate.
2. In Figure 2(e), there is no X axis title; it is a bit difficult to understand. Please reorganize this figure.
3. In Figure 3a, please enlarge the note of ID and IG for clear presentation.
4. The authors mentioned that the shear exfoliation is efficient than sonication. And the A2B3 crystals such as Sb₂S₃ could be exfoliated via high shear exfoliation. However, from context and the supporting information, barely no information focuses on the working mechanism of shear exfoliation and the devices they used. Please provide more description/discussion for better understanding.
5. In addition, it should be specified whether the discharge capacity is calculated based on the total mass (Sb₂S₃ and carbon) or the weight of active material Sb₂S₃.
6. In Figure 4, the first discharge capacity for the SBS and SBS/C electrode is very high, nearly 1200 mAh g⁻¹, which is higher than the theoretical capacity. Please give explanation. And why is the first coulombic efficiency so low?
7. Although few-layered SBS/C anode shows good performance in K ion batteries, this manuscript does not provide some important parameters of the electrodes, such as loading amount of active materials. Without these parameters, it is not fair to compare the performance of the electrode with those of the other electrodes.
8. In Figure 4 (h), the SBS/C (16:1 and 32:1) electrodes exhibit a slightly higher capacity at 50mA/g after rate test, compared with other electrodes. The authors are encouraged to discuss possible reasons for this phenomenon.

Reviewer #2 (Remarks to the Author):

This paper reports that the few-layered SBS/C composite was prepared by one-step high-shear exfoliation in ethanol/water solvent. The authors used the few-layered SBS/C composite as anode of potassium ion battery, achieving outstanding performance with high reversible capacity, good rate capability, long cycling stability. The work is interesting and this manuscript is logically organized well. However, this manuscript needs major revisions before acceptance, due to the following reasons:

- (1) The authors claim that the obtained materials are few-layered Sb₂S₃/C. However, from all TEM and SEM, it is not obvious. Please provide more clear evidence.

(2) The first charge-discharge curve is very important to explain its electrochemical properties. Please give the first charge-discharge data in Fig. 4 e.

(3) The carbon in the composite can greatly affect the electrochemical properties of the battery, the authors should make sure a similar mass content of Carbon in these SBS/C composites as far as possible. Otherwise, this comparison makes no sense.

(4) Whether the good electrochemical performance was caused by different carbon content or because of the different layers, the reason was unknown. The author should make more in-depth comparison and discussion.

(5) In figure 4b and d, discharge capacity of the ordinate is set to -200mAhg⁻¹, which is suspected to be intentionally misleading readers. Please correct it.

(6) The quality of Figure 2c should be improved. Its axis is blurred, please adjust it.

(7) As we known, the discharge capacity at lower current density often is higher than that of high current density. However, Fig. 4,b,d, h shows an opposite result. The authors should further explain the unusual phenomena.

(8) If possible, the long cycle performances should be further investigated by the authors to show the better stability of SBS/C.

Reviewer #3 (Remarks to the Author):

The manuscript of Y. Liu et al. describes a facile synthesis of a few layered antimony sulphide/carbon sheet (SBS/C) composite and its electrochemical performance as anode for potassium ion batteries. Unlike sodium ion batteries that have attracted significant attention from the research community in the last five years, potassium-ion cells are in an earlier stage of evolution, although the interest in these systems is clearly increasing. Authors describe the novel and interesting results on the preparation of the anode material via simple one-step high-shear exfoliation in ethanol/water solvent and its superior cycling stability, rate capability and impressive electrochemical capacity (above 400 mAh/g after 200 cycles at current density of 500mA/g). All described characteristics of the material make it attractive and promising as an active component of the anode in potassium ion batteries. Results presented by Y. Liu et al. are reliable, fully consistent and highly interesting in many relations, including the studies of mechanism of the exfoliation and potassium interaction with anode material during the cycling. Presented results and conclusions will be of interest to others in the community and the wider field.

However, some points have to be clarified and additional data should be provided before I recommend this manuscript for publication in Nature Communications.

1. Authors suggest the formation of the Sb and K₂S₆ phases during the potassium intercalation to the anode based on the XRD data presented on the Figure 1a. However, the corresponding peaks have a very low intensity, and it seems impossible to distinguish them from the noise of the diffractogram. This point needs to be clarified and discussed, and probably additional frame with a higher resolution should be presented in Figure 1.

2. In order to gain insight on the chemical composition and structure of the carbon in SBS/C composite the additional XPS studies (carbon 1s peak) should be provided. Additionally, the probable oxidation (or reduction) of the antimony during the high-shear exfoliation in ethanol/water solvent and formation of the antimony (III or V) oxides (or elemental antimony) on the surface of the Sb₂S₃ nanosheets should be discussed based on the XPS (Sb 3d peak) studies.

3. The cyclic voltammograms were assigned based on in-situ XRD analysis and batteries research on SBS anode in lithium and sodium ion batteries (page 16). However, the presented references

12 and 20 correspond to the Sb₂S₃ studies as sodium ion batteries anodes. I would recommend to cite additionally the article of P.V. Prikhodchenko et al. (Chemistry of Materials, 2012, 24, pp 4750–4757) describing the Sb₂S₃@Graphene for a superior lithium battery anode.

4. The elemental sulfur and SO₂ content in the SBS/C composite are discussed on the page 17 but all values are presented only in ESI. Some conclusions like " it should be noted that the element sulfur was detected in both electrodes and it shows much higher intensity/content in SBS/C than bulk SBS" should be supported with values of the sulfur content directly in the manuscript.

5. The similar comment could be made regarding the discussion on the page 19 on the carbon content in the SBS/C composites. Values of the carbon content should be introduced into the text of the manuscript.

6. I would suggest to revise the structure of the manuscript. The discussion part of the manuscript is in fact conclusions, but the chapter "results" actually contains "results and discussion".

Response letter for the manuscript entitled “Boosting potassium ion battery performance by few-layered Sb_2S_3 /carbon anode prepared via solution-triggered one-step shear exfoliation” (NCOMMS-18-05306A)

Response to Reviewers:

We have carefully considered all the comments and questions raised by the reviewers. We took time to plan and carry out additional experiments, which helped to address the reviewers' comments and questions. Newly obtained data are included in the manuscript or supplementary information and the relevant discussions have been amended in the manuscript. One of major changes in the manuscript is related to the discussion of the chemical composition of SBS after exfoliation and the structure of the carbon in SBS/C composite with the support of additional XPS data (Fig. 3c, Supplementary Fig. 12). Another major revision in the manuscript focuses on the in-depth comparison and discussion of the effects of the carbon content and the thickness of SBS on the electrochemical performance with support of an additional experiment (Supplementary Fig. 20). In order to clearly present the morphology of the layered SBS/C composite, we have re-conducted TEM, and the collected TEM images are shown in Fig. 3e and Supplementary Fig. 13. Additionally, we have explained the unusual phenomena in the rate performance as suggested by Reviewer #2 and re-conducted rate performance tests by extending the standing time up to 24 hrs before the test (Supplementary Fig. 18). Moreover, the long-term cycling performance of SBS/C composite was investigated as suggested (Supplementary Fig. 24). We kindly thank the reviewers for raising relevant questions and for constructive comments which have, in our opinion, helped us to improve the quality of the present work.

Point by Point Responses:

Reviewer #1: This manuscript reported the high-performance few-layered Sb_2S_3 /C composite anode for potassium ion battery, which is fabricated via novel technique of one-step shear exfoliation. The well-developed few-layered structure of SBS anode enhanced the potassium ion battery performance, with capacity superior to the traditional graphite anode for LIBs. Based on the abundance and the low cost of potassium than lithium, this system could be promising alternative for LIBs. Besides, the achievement and new findings regarding the in-situ carbonization during exfoliation and the small amount of S production during

electrochemical cycling in potassium ion battery are all impressive and interesting, which could be beneficial for structure design and deep mechanism understanding. Overall, this manuscript is well organized and written. And the experimental aspects of the research are thoroughly detailed and well explained. I recommend its publication after addressing the following comments and I am sure the following revisions would serve to further strengthen the manuscript.

Q1. Please label the figures in sequence according to the description in context. For example, description of Figure S2i in advance of Figure S2d-g is not appropriate.

A: Thanks for pointing this out. According to your suggestion, we have relabelled the Supplementary Fig. 3 (Figure S2 in the previous version). In manuscript, we changed the previous Figure S2i to **Supplementary Fig. 3d** and changed the previous Figure 2d-g to **Fig. 3e-h** (highlighted in manuscript in red).

Q2. In Figure 2(e), there is no X axis title; it is a bit difficult to understand. Please reorganize this figure.

A: Thanks for the suggestion. We have revised Fig. 2e as shown below:

Fig. 2 (e) Adsorption energy of water and ethanol molecules on Sb₂S₃ (010) surface;

Q3. In Figure 3a, please enlarge the note of ID and IG for clear presentation.

A: Thanks for the suggestion. The labelling of I_D and I_G has been enlarged, as shown below:

Fig. 3 (a) Raman spectra of the exfoliated products in different solvents, SBS (W), SBS/C (E), and SBS/C (E/W = 2:1) compared with commercial SBS, indicating the generation of amorphous carbon during exfoliation in ethanol-containing solvents.

Q4. The authors mentioned that the shear exfoliation is efficient than sonication. And the A_2B_3 crystals such as Sb_2S_3 could be exfoliated via high shear exfoliation. However, from context and the supporting information, barely no information focus on the working mechanism of shear exfoliation and the devices they used. Please provide more description/discussion for better understanding.

A: Thanks for your advice. As introduced in the experimental section, the mixer used was a L5M high shear laboratory mixer, made by Silverson Machines Ltd., UK. A typical features of the mixing head (the interchangeable, screw-on slotted disintegrating head) is the narrow gap between its rotor and the screen, ~ 0.05 mm, as shown in Supplementary Fig. 25. When it is in operation, there is a high rotor speed (N) creating a high shear rate ($\dot{\gamma}$) within the gap¹. In the whole exfoliation process, both mechanical shearing and cavitation are responsible for the evolution of the size/thickness of the particles². What is more, based on our research on the solvent effect of water, that water can be easily intercalated into the open channels of SBS, and fracturing/exfoliation could occur more easily owing to the accumulated stress in the SBS crystal and the outside mechanical shear force.

1. Paton, K. R., Varrla, E., Backes, C., Smith, R. J., Khan, U., Neill, A. O', Boland, C., Lotya, M., Istrate, O. M., King, P., Higgins, T., Barwich, S., May, P., Puczkarski, P., Ahmed, I., Moebius, M., Pettersson, H., Long, E., Coelho, J., O'Brien, S. E., McGuire, E. K., Sanchez, B. M., Duesberg, G. S., McEvoy, N., Pennycook, T. J., Downing, C., Crossley, A., Nicolosi, V.,

Coleman, J. N. Scalable production of large quantities of defect-free few-layer graphene by shear exfoliation in liquids. *Nature Materials* **13**, 624-630 (2014).

- Tai, Z., Subramaniam, C. M., Chou, S. L., Chen, L., Liu, H. K., Dou, S. X. Few atomic layered lithium cathode materials to achieve ultrahigh rate capability in lithium-ion batteries. *Adv. Mater.* **29**, 1700605 (2017).

A schematic illustration of the mixing head of mixer has been added to the Supporting Information, and related explanation has been added to the revised manuscript as shown below:

Supplementary Fig. 25 Mixing head of exfoliation machine. Bottom view (left) and flat view (right) of mixing head with indicated rotor and stator.

“The mixer used was an L5M high shear laboratory mixer, made by Silverson Machines Ltd., UK. Here, the interchangeable, screw-on slotted disintegrating head has a rotor 30 mm in diameter and the gap between its rotor and the screen is ~0.05 mm (illustrated in Supplementary Fig. 25). When it is in operation, the high rotor speed (N) creates a high shear rate ($\dot{\gamma}$) within the gap.” (Experimental Section, page 17, line 9)

Q5. In addition, it should be specified whether the discharge capacity is calculated based on the total mass (Sb_2S_3 and carbon) or the weight of active material Sb_2S_3 .

A: Thanks for your suggestion. In my work, the capacity is calculated based on the total mass of the active materials (the mass including SBS and carbon for SBS/C electrode and the mass of SBS for SBS electrode). The relevant description has been added to the revised manuscript as follows:

“All the capacities of SBS/C electrodes were calculated based on the total mass of SBS and carbon.” (Page 11, line 19)

Q6. In Figure 4, the first discharge capacity for the SBS and SBS/C electrode is very high, nearly 1200 mAh g⁻¹, which is higher than the theoretical capacity. Please give explanation. And why is the first coulombic efficiency so low?

A: Thanks for your comment. The first discharge capacity of the SBS and SBS/C electrode is very high, due to the solid electrolyte interphase (SEI) formation and side reactions between electrode and electrolyte. As is well known, the initial coulombic efficiency (ICE) for commercial anode electrode in LIBs normally exceeds 90% in order to successfully pair the cathode material to maintain the cycling. However, the ICE is normally quite low for nanostructured electrode¹⁻². The first cycle irreversible capacities are mainly due to the formation of solid electrolyte interphase (SEI). For the nanostructured few-layered SBS or SBS/C composite, the largely exposed surface and the defects formed during exfoliation results in side reactions, which lead to the low coulombic efficiency.

According to the reported literature, the low initial coulombic efficiency could be overcome and improved via pre-lithiation, the use of ether-based electrolyte, and employing active materials with a secondary structure³⁻⁵. Further systematic research on improving the ICE and full-cell engineering will be conducted in the future in order to facilitate the commercial application of SBS in KIBs.

1. Yang, J., Ju, Z., Jiang, Y., Xing, Z., Xi, B., Feng, J., Xiong, S. Enhanced capacity and rate capability of nitrogen/oxygen dual-doped hard carbon in capacitive potassium-ion storage. *Adv. Mater.* **30**, 1700104 (2018).
2. Zhang, W., Mao, J., Li, S., Chen, Z., Guo, Z. Phosphorus-based alloy materials for advanced potassium-ion battery anode. *J. Am. Chem. Soc.* **139**, 3316-3319 (2017).
3. Liu, N., Hu, L., McDowell, M. T., Jackson, A., Cui, Y. Prelithiated Silicon Nanowires as an Anode for Lithium Ion Batteries. *ACS Nano*, **5**, 6487 (2011).
4. Lei, K., Li, F., Mu, C., Wang, J., Zhao, Q., Chen, C., Chen, J. High K-storage performance based on the synergy of dipotassium terephthalate and ether-based electrolytes. *Energy Environ. Sci.* **10**, 552-557 (2017).
5. Liu, N., Lu, Z., Zhao, J., McDowell, M. T., Lee, H.-W., Zhao, W., Cui, Y. A pomegranate-inspired nanoscale design for large-volume-change lithium battery anodes. *Nat. Nanotechnol.* **9**, 187 (2014).

Q7. Although few-layered SBS/C anode shows good performance in K ion batteries, this manuscript does not provide some important parameters of the electrodes, such as loading

amount of active materials. Without these parameters, it is not fair to compare the performance of the electrode with those of the other electrodes.

A: Thanks for your suggestion. The loading mass of the active materials (SBS and SBS/C) in my work is around 1 mg cm^{-2} . The related description has been added to *Electrochemical Measurements* in the **Experimental Section** as follows:

“The loading mass of the active materials (SBS or SBS/C) was around 1 mg cm^{-2} .” (Page 18, line 23)

Q8. In Figure 4 (h), the SBS/C (16:1 and 32:1) electrodes exhibit a slightly higher capacity at 50 mA/g after rate test, compared with other electrodes. The authors are encouraged to discuss possible reasons for this phenomenon.

A: Thanks for your suggestion. We believe that the slightly higher capacities for SBS/C (16:1 and 32:1) at 50 mA g^{-1} after the rate tests are caused by the higher carbon content in the SBS/C composites compared to the other SBS/C samples (1:1 to 8:1). In order to fabricate few-layered SBS/C electrode, the mixed solvent of water/ethanol was exploited for shear exfoliation of bulk SBS. Fig. 4h shows the rate performance and the further cycling performance of electrode exfoliated in different ratios of ethanol/water solution ($E/W = 1:1$, $E/W = 2:1$, $E/W = 6:1$, $E/W = 8:1$, $E/W = 16:1$, $E/W = 32:1$). Based on the EDS results, it was found that with increasing ethanol content in the exfoliation solution, the carbon content in the composites increases. A slightly higher carbon content compared to other samples in these electrodes plays an important role in electron transfer with improved electrical conductivity.

Reviewer #2: This paper reports that the few layered SBS/C composite was prepared by one-step high-shear exfoliation in ethanol/water solvent. The authors used the few-layered SBS/C composite as anode of potassium ion battery, achieving outstanding performance with high reversible capacity, good rate capability, long cycling stability. The work is interesting and this manuscript is logically-organized well. However, this manuscript needs major revisions before acceptance, due to the following reasons:

Q1. The authors claim that the obtained materials are few-layered $\text{Sb}_2\text{S}_3/\text{C}$. However, from all TEM and SEM, it is no obvious. Please provide more clear evidence.

A: Thanks for your comment. Your suggestion is valuable and we have already tried our best to provide clear TEM images with few-layered Sb_2S_3 and carbon layers. Due to the thickness difference, we can clearly observe and identify the layered SBS and the carbon sheets separately from the TEM images. However, it is difficult to focus on both of them at the same time. If we focus on the carbon, then the SBS sheets seem to blur, and if we focus on the SBS sheets the carbon sheet are not clearly present. This limitation has affected the quality of the TEM images of the samples. So, it is very difficult to acquire a good TEM image with the layered SBS and carbon together. We have redone TEM, and the images of few-layered $\text{Sb}_2\text{S}_3/\text{C}$ are provided below (Fig. 3e and Supplementary Fig. 13). From the TEM images, the size of SBS sheets is a few hundred nanometers, and the thickness of the sheets is only a few nanometers (as shown in AFM images in Supplementary Fig. 13e and Fig. 2c). We believe the newly obtained TEM images and the AFM images are clear enough to show the few-layered nanosheets.

In the revised manuscript, we have updated TEM image in the Fig. 3e, and also provided more images in Supplementary Fig.13 in the Supporting Information as shown below.

Supplementary Fig. 13 Morphology of as-prepared layered SBS/C (E/W=2:1) composite. (a-d) TEM images of SBS/C composite. (e) Typical AFM images with cross-sectional height profiles of SBS nanosheets.

Fig. 3 (e) TEM image of the exfoliated SBS/C composite (E/W = 2:1).

Q2. The first charge-discharge curve is very important to explain its electrochemical properties. Please give the first charge-discharge data in Fig. 4 e.

A: Thanks for your suggestion. The first charge-discharge curves have been added to the Fig. 4e in the revised manuscript. The first discharge curve has obvious plateaus, and the capacity

is very high compared with the following cycles mainly due to the SEI formation and possibly other side reactions between the electrode and the electrolyte. By combining these results with in-situ synchrotron XRD data and CV results, we believe that the intercalation initiated at around 1.1 V during the first discharge, then the conversion reaction occurred at the voltage plateau of 0.6 V, followed by the alloying reaction at around 0.4 V. The first charge curve almost overlaps with the second charge curve.

Fig. 4 (e) Discharge/charge curves at different cycles for SBS/C (NS-2-E) composite electrode at 50 mA g⁻¹.

Q3. The carbon in the composite can greatly affect the electrochemical properties of the battery, the authors should make sure a similar mass **content of carbon** in these SBS/C composites as far as possible. Otherwise, this comparison makes no sense.

Q4. Whether the good electrochemical performance was caused by **different carbon content** or because of the **different layers**, the reason was unknown. The author should make more in-depth comparison and discussion.

A for Q3 and Q4: Thanks for your valuable comments and suggestion. These concerns are thought-provoking, which can inspire us to deeply consider and study the influence of the nanosheet thickness and carbon content in the composites on their electrochemical

performance. The SBS/C composites fabricated from one-step exfoliation have two variables, one is the thickness of SBS and the other is the carbon content in composite.

In order to investigate how each factor (different thickness and carbon content) affects the electrochemical performance, we conducted **two-step exfoliation** and obtained samples with different thicknesses first, then chose the best thickness and incorporated carbon into the sample with various content.

First of all, we fabricated SBS with different thickness in water by adjusting the rotation rate (4000, 5000, 6000, 7000 rpm). The electrochemical performance was investigated for the samples of SBS 4000, SBS 5000, SBS 6000 and SBS 7000 samples in order to understand how the thickness of SBS nanosheets affects the electrochemical performance. Then, we choose exfoliated SBS-6000 as the target sample and add different amounts of ethanol (1:1, 4:1, 16:1) for the step-two exfoliation. In this way, we tried to understand the influence of the carbon content on the electrochemical performance of SBS/C with the same thickness of layered SBS. The collected data and relevant discussion are shown below.

Supplementary Fig. 20 a-b shows the cycling stability and rate capability of layered SBS 4000, SBS 5000, SBS 6000, and SBS 7000 electrodes. With the rotation rate increasing, the thickness of the SBS nanosheets decreased, as shown in Supplementary Fig. 20c. The cycling stability and rate performance improved with the decrease of the thickness of the SBS and among them, the SBS-6000 electrode shows the best electrochemical performance. The relatively poor cycling stability of SBS-7000 may be due to the increased defects from the deep exfoliation.

After introducing the carbon sheets via the step-two exfoliation, the electrochemical performance (rate capability and cycling stability) was significantly enhanced compared with that of pure few-layered SBS electrodes (Supplementary Fig. 20). For the SBS/C composite exfoliated from the two-step exfoliation (with the SBS thickness the same in all composites), the influence of carbon content on the electrochemical performance seems to not be very significant. The capacities of electrodes with various carbon contents almost overlap with each other at different current densities (Supplementary Fig. 20d).

In conclusion, the thickness (in the range of 5-30nm) of SBS sheets dramatically influenced the cycling stability and rate capability of SBS electrodes. A small weight percentage of carbon sheets in the SBS/C could improve the cycling stability and rate capability significantly compared with pure SBS electrodes, however, carbon content in the range of 3.5%-5% plays a negligible role in the electrochemical performance, especially the rate capabilities. In this case,

when we go back to scrutinize the electrochemical performance of SBS/C composites fabricated via one-step exfoliation, we could conclude that the electrochemical performance of SBS/C composite electrodes is dependent on both the thickness of SBS and content of the carbon, however, the differences in electrochemical performance between the various SBS/C electrodes were mainly caused by the thickness of SBS, possibly due to the narrow range of carbon content (3.5-5 wt%).

Supplementary Figure 20 has been added into the Supporting Information, and relevant descriptions/discussions have been added in the revised manuscript as follows.

“In order to distinguish the influence of carbon content and the thickness of SBS sheets on the electrochemical performance of composite electrodes, we designed and fabricated SBS/C composites via a two-step shear exfoliation. It was found that the cycling stability and rate performance improved with decreasing thickness of the SBS sheets, and among them the SBS-6000 electrode shows the best electrochemical performance. We then investigated the influence of the carbon content based on the same thickness of SBS. Supplementary Fig. 20 suggests that although the presence of carbon sheets improved the cycling stability and rate capability significantly, the carbon content in the composites (in the range from 3.5 wt % to 5 wt %) plays a negligible role in influencing the electrochemical performance. Therefore, the differences in electrochemical performance of the few-layered SBS/C electrodes shown in Fig. 4h are mainly caused by the thickness of SBS.”(Page 14, line 8)

Supplementary Fig. 20 Electrochemical properties of few-layered SBS and SBS/C electrodes fabricated via two-step exfoliation. Comparison of (a) cycling performance and (b) rate performance of SBS 4000, SBS 5000, SBS 6000, and SBS 7000 electrodes, which were exfoliated in water with different rotation rates. (c) Thickness distributions for SBS 4000, SBS 5000, SBS 6000, and SBS 7000 electrodes. Comparison of (d) cycling performance and (e) rate performance of SBS/C (E/W = 1:1), SBS/C (E/W = 4:1), and SBS/C (E/W = 16:1) electrodes, which were exfoliated via two-step exfoliation (based on the exfoliated SBS 6000 products, with different weight ratios of ethanol added for the second exfoliation). (e) Carbon weight percentages of the SBS/C (E/W = 1:1), SBS/C (E/W = 4:1), and SBS/C (E/W = 16:1) samples.

Experimental Section

“For the two-step exfoliation, different thicknesses of SBS in water were fabricated with adjusting the rotation rate (4000, 5000, 6000, 7000 rpm) (SBS 4000, SBS 5000, SBS 6000, SBS 7000). Then, the solution product of SBS 6000 was chosen as target sample, and different amounts of ethanol (ethanol/water ratio: 1:1, 4:1, 16:1) were added for step-two exfoliation.” (Page 18, line 1)

Q5. In figure 4b and d, discharge capacity of the ordinate is set to -200mAhg^{-1} , which is suspected to be intentionally misleading readers. Please correct it.

A: Thanks for your reminder. We have reset the ordinate to 0mAhg^{-1} as shown below.

Fig. 4 Comparison of (a) cycling performance and (b) rate performance of SBS/C (NS-1-E), SBS/C (NS-2-E), and SBS/C (NS-3-E) electrodes, which were exfoliated and collected in ethanol. Comparison of (c) cycling performance and (d) rate performance of layered SBS (NS-1-W), SBS (NS-2-W), and SBS (NS-3-W) electrodes, which were exfoliated and collected in water.

Q6. The quality of Figure 2c should be improved. Its axis is blurred, please adjust it.

A: Thanks for your advice. We have adjusted the axis to make it clear, as shown in Figure below.

Fig. 2 (c) Histograms of the thickness distribution of SBS nanosheets prepared with different solvents (mixed ethanol and water (E/W = 2:1), pure ethanol (E), and pure water (W)), and typical AFM images with inset cross-sectional height profiles of SBS nanosheets obtained from SBS/C (E/W = 2:1, c1) and SBS (W, c2).

Q7. As we known, the discharge capacity at lower current density often is higher than that of high current density. However, Fig. 4,b,d, h shows an opposite result. The authors should further explain the unusual phenomena.

A: Thanks for your comment and careful checking. We all know that the discharge capacity is normally higher at a lower current density than at a higher current density. In our case, however, the discharge capacities at current density of 20 mA g⁻¹ are lower than at current density of 50 mA g⁻¹ (Fig. 4b, d) and the initial discharge capacities at 50 mA g⁻¹ are lower than at the current density of 100 mA g⁻¹ (Fig. 4h). We believe that this phenomena can be ascribed to an activation process.

In order to support this hypothesis, we assembled coin-cells and retested the rate performance in the following two ways using the electrode of SBS/C (E/W=4:1) electrode (mentioned in Q3 and Q4). We first reset the procedure for rate testing (20, 50, 100, 200, 20, 300, 500 mA g⁻¹, 1 A g⁻¹) to see if the capacities changed. In the initial cycles, part of active materials may not fully participate in the electrochemical reaction, so that the capacity at 20 mA g⁻¹ is low, however, as can be seen from Supplementary Fig. 18a, when the current density returns to 20 mA g⁻¹ after 20 cycles, the capacities at 20 mA g⁻¹ are slightly higher than the discharge

capacities at 50 mA g⁻¹, suggesting that unusual phenomenon only occurred in the first few cycles.

We also tested the rate performance of SBS/C (E/W = 4:1) electrode after different standing times (6 h, 12 h, 24 h) (Supplementary Fig. 18b). It can be observed that the initial discharge capacities increase with increasing standing time. The discharge capacity at 20 mA g⁻¹ is almost same as that at 50 mA g⁻¹ when the standing time is 24 hours, demonstrating that the unusual phenomenon may due to the activation process in the initial cycles and that increasing the standing time of fresh cells will let the electrolyte effectively soak into the electrode, thus improving the utilization of active materials in the initial cycles. This activation phenomenon was also reported for other anode and cathode materials¹⁻².

- 1 Szczech, J. R., Jin, S., Nanostructured silicon for high capacity lithium battery anodes. *Energy Environ. Sci.* **4**, 56-72 (2011)
- 2 Mao, Y., Li, G., Guo, Y., Li, Z., Liang, C., Peng, X., Lin, Z. Foldable interpenetrated metal-organic frameworks/carbon nanotubes thin film for lithium–sulfur batteries. *Nat. Commun.* **8**, 14628 (2017)

We have made some revisions to the manuscript and added Supplementary Fig. 18 to the Supporting Information as shown below.

“It was also found that the initial discharge capacities of SBS/C electrodes at 50 mA g⁻¹ is slightly lower than the following capacities at 100 mA g⁻¹ (Fig. 4h) in the rate test, similar unusual phenomena is also observed in Fig. 4b, d, which can be ascribed to the activation process of SBS and SBS/C electrodes in the initial cycles and could be avoided by extending the stand time of fresh cells (Supplementary Fig. 18).” (Page13, line 25)

Supplementary Fig. 18 Rate test under certain settings and with different standing times before the test. (a) Rate performance of SBS/C (E/W = 4:1) with standing time of 12 h before the test (at the current densities of 20, 50, 100, 200, 20, 300, 500 mA g⁻¹, and 1 A g⁻¹). (b) Rate performance of SBS/C (E/W = 4:1) electrodes with different standing times (6 h, 12 h, 24 h).

Q8. If possible, the long cycle performances should be further investigated by the authors to show the better stability of SBS/C.

A: Thanks for your advice. The long-term cycling performance of SBS/C (E/W=2:1) electrode was investigated at a current density of 1A g⁻¹ (Supplementary Fig. 24). The relevant revisions to the manuscript and supplementary Fig. 24 are shown below:

“The long-term cycling performance of SBS/C (E/W = 2:1) electrode was further investigated as shown in Supplementary Fig. 24. It shows excellent cycling stability and high capacity retention of 79% after 1000 cycles at a current density of 1 A g⁻¹.” (Page 15, line 3)

Supplementary Fig. 24 Cycling stability of SBS/C (E/W=2:1) electrode. Long-term cycling performance of SBS/C (E/W = 2:1) electrode at the current density of 1 A g^{-1}

Reviewer #3: The manuscript of Y. Liu et al. describes a facile synthesis of a few layered antimony sulphide/carbon sheet (SBS/C) composite and its electrochemical performance as anode for potassium ion batteries. Unlike sodium ion batteries that have attracted significant attention from the research community in the last five years, potassium-ion cells are in an earlier stage of evolution, although the interest in these systems is clearly increasing. Authors describe the novel and interesting results on the preparation of the anode material via simple one-step high-shear exfoliation in ethanol/water solvent and its superior cycling stability, rate capability and impressive electrochemical capacity (above 400 mAh/g after 200 cycles at current density of 500 mA/g). All described characteristics of the material make it attractive and promising as an active component of the anode in potassium ion batteries. Results presented by Y. Liu et al. are reliable, fully consistent and highly interesting in many relations, including the studies of mechanism of the exfoliation and potassium interaction with anode material during the cycling. Presented results and conclusions will be of interest to others in the community and the wider field.

However, some points have to be clarified and additional data should be provided before I recommend this manuscript for publication in Nature Communications.

Q1. Authors suggest the formation of the Sb and K_2S_6 phases during the potassium intercalation to the anode based on the XRD data presented on the Figure 1a. However, the corresponding peaks have a very low intensity, and it seems impossible to distinguish them from the noise of the diffractogram. This point needs to be clarified and discussed, and probably additional frame with a higher resolution should be presented in Figure 1.

A: Thanks for your suggestion. The newly formed discharge intermediate products after conversion and alloying reaction are nano-crystalline particles (Supplementary Fig. 2) which makes weak and broad diffraction signals. That nano-sized material caused low-intensity or the disappearance of peaks in XRD has been proposed, according to the reported literature.¹

In order to provide additional evidence for the intermediate products (Sb and K₂S₆ phases) during discharge, we fabricated the cells again and conducted the ex-situ SAED on the electrode after discharged to 0.5 V. As shown, the SAED pattern was composed of several weak Debye–Scherrer concentric rings, superimposed with some discrete Laue spots, and a certain number of bright Laue spots, as there are insufficient numbers of crystal planes oriented in all directions, within the small selected area. The diffused green rings with bright spots indicate the poly-crystallinity but nano-size natures and can be assigned to (040) and (002) planes of K₂S₆ phase. The K₂S₆ (040) corresponds to the reflection at 13.4° in synchrotron XRD pattern. On the other hand, only bright spots, corresponding to (101)/(002) planes and the weak peaks (15.17/15.68°) in the synchrotron XRD pattern, are observed for metallic Sb phase.

1. Poizot, P., Laruelle, S., Grugeon, S., Dupont, L., Tarascon, J.M. Nano-sized transition-metal oxides as negative-electrode materials for lithium-ion batteries. *Nature* **407**, 496-499 (2000).

The relevant discussion and revisions have been added to the manuscript, and Fig. 1a was improved with additional SAED additional data.

“Due to the nano-crystallinity of the intermediate products of Sb and K₂S₆ (Supplementary Fig. 2), the corresponding diffraction peaks are weak and broad. In order to confirm the existence of intermediate products, ex-situ SAED was conducted on the electrode after it was discharged to 0.5 V. As shown in the SAED pattern (Fig. 1a), the marked spots in orange are corresponding to the Sb (101) and (002) planes, which is consistent with the weak peaks (15.17 and 15.68°) in the synchrotron XRD pattern, and the diffused green rings with spots belong to the K₂S₆ (040), which corresponds to the peak at 13.4° in synchrotron XRD.” (Page 6, line 23)

Fig. 1 (a) In-situ synchrotron XRD patterns of Sb_2S_3 electrodes upon K insertion at various potentials (left) and ex-situ SAED pattern (right) at 0.5 V with high-resolution image revealing weak reflections.

Supplementary Fig. 2 Morphology of commercial Sb_2S_3 after discharge to 0.5 V. TEM image of commercial Sb_2S_3 after discharge to 0.5 V with the yellow circle indicating the selected area for SAED in Fig. 1a.

Q2. In order to gain insight on the chemical composition and structure of the carbon in SBS/C composite the additional XPS studies (carbon 1s peak) should be provided. Additionally, the probable oxidation (or reduction) of the antimony during the high-shear exfoliation in ethanol/water solvent and formation of the antimony (III or V) oxides (or elemental antimony) on the surface of the Sb_2S_3 nanosheets should be discussed based on the XPS (Sb 3d peak) studies.

A: Thanks for your advice. Additional XPS analysis on the C 1s (Fig. 3c and Supplementary Fig. 12d) and Sb_{3d} peaks (Supplementary Fig. 12a-c) for the SBS/C (E/W = 2:1), SBS/C (E) and SBS (W) samples has been provided in the revised manuscript and Supporting Information, and the relevant discussion has been amended in the revised manuscript. From Supplementary Fig. 12a-c (shown below), the peaks at 539.1 eV and 529.7 eV correspond to the binding energy of Sb 3d_{3/2} and Sb 3d_{5/2}^{1,2,3}, respectively and the peak splitting of 9.4 eV is consistent with that for Sb in the oxidation state of +3. Two extra peaks at 531 eV and 540.4 eV for the SBS (W) can be found and are ascribed to the chemical state of Sb₂O₃², indicating partially oxidation of Sb₂S₃ nanosheets exfoliated in water. While, as for samples of SBS/C exfoliated in ethanol and ethanol/water solution, however, the fitted peaks corresponding to Sb₂O₃ cannot be found, suggesting that the oxide phase was not formed on the surface of Sb₂S₃ and that a relatively high-purity Sb₂S₃ phase is produced when it is exfoliated in ethanol-containing solvent. The C 1s XPS profile at the surface was also obtained and deconvoluted as well in order to understand the composition and structure of carbon in the SBS/C composites. Peaks due to C=C (285.3 eV) and C-C (284.4 eV) are evident (Fig. 3c)⁴; these two bonds account for around 90 % of total carbon bonds based on the XPS analysis. The intensities of the peaks of C=O and C-O (286.5 eV), and of COOR (288.4 eV)⁵ are slightly higher for SBS/C exfoliated in ethanol/water solution than for that exfoliated in pure ethanol, and these functional groups evidenced in the C 1s spectrum are consistent with the FTIR results (Fig. 3b).

- 1 Choi, Y. C., Seok, S., Efficient Sb₂S₃-sensitized solar cells via single-step deposition of Sb₂S₃ using S/Sb-ratio-controlled SbCl₃-thiourea complex solution. *Adv. Funct. Mater.* **25**, 2892–2898 (2015)
- 2 Kim, D. H., Lee, S. J., Park, M. S., Kang, J. K., Heo, J. H., Im, S. H., Sung, S. J. Highly reproducible planar Sb₂S₃-sensitized solar cells based on atomic layer deposition. *Nanoscale* **6**, 14549 (2014)
- 3 Prikhodchenko, P. V., Gun, J., Sladkevich, S., Mikhaylov, A. A., Lev, O. L., Tay, Y. Y., Batabyal, S. K., Yu, D. Y. W., Conversion of hydroperoxoantimonate coated graphenes to Sb₂S₃@graphene for a superior lithium battery anode. *Chem. Mater.* **24**, 4750-4757 (2012).
- 4 Lee, E., Lee, S. G., Lee, H. C., Jo, M., Yoo, M. S., Cho, K. Direct growth of highly stable patterned graphene on dielectric insulators using a surface-adhered solid carbon source. *Adv. Mater.* **30**, 1706569 (2018).

- 5 Picot, O. T., Rocha, V. G., Ferraro, C., Ni, N., D'Elia, E., Meille, S., Chevalier, J., Saunders, T., Peijs, T., Reece, M. J., Saiz, E. Using graphene networks to build bioinspired self-monitoring ceramics. *Nat. Commun.* **8**, 14425 (2017).

We added Fig. 3c and relevant discussions in the revised manuscript and added Supplementary Fig. 12 in the Supporting Information as shown below:

“The Sb 3d and C 1s XPS profiles of SBS/C and SBS were also obtained and deconvoluted in order to understand the composition and structure of the SBS/C composite. It was noted that the chemical state of Sb_2O_3 could be found for SBS exfoliated in water, with the peaks at 531 eV and 540.4 eV²⁷, indicating partial oxidation of Sb_2S_3 nanosheets, while for samples of SBS/C exfoliated in ethanol and ethanol/water solutions, these two peaks corresponding to Sb_2O_3 are absent (only 539.1 eV and 529.7 eV for Sb 3d_{3/2} and Sb 3d_{5/2} of Sb_2S_3)¹³, suggesting that the oxide phase was not formed on the surface of Sb_2S_3 and that a relatively high-purity Sb_2S_3 phase can be produced when exfoliated in ethanol-containing solvent (Supplementary Fig. 12a-c). As for C 1s profile, peaks due to C=C (285.3 eV) and C-C (284.4 eV) are evident (Fig. 3c and Supplementary Fig. 12d)²⁸; these bonds are mainly from the construction of carbon in composite, and the content is calculated to be about 90%. The intensities of the peaks of C=O and C-O (286.5 eV), and of COOR (288.4 eV)²⁹ are slightly higher for SBS/C exfoliated in ethanol/water solution than for its counterpart exfoliated in pure ethanol, and these functional groups evidenced in the C 1s spectrum are consistent with the FTIR results shown in Fig. 2b.”
(Page 10, line 20)

- 27 Kim, D. H., Lee, S. J., Park, M. S., Kang, J. K., Heo, J. H., Im, S. H., Sung, S. J. Highly reproducible planar Sb_2S_3 -sensitized solar cells based on atomic layer deposition. *Nanoscale* **6**, 14549 (2014)
- 28 Lee, E., Lee, S. G., Lee, H. C., Jo, M., Yoo, M. S., Cho, K. Direct growth of highly stable patterned graphene on dielectric insulators using a surface-adhered solid carbon source. *Adv. Mater.* **30**, 1706569 (2018).
- 29 Picot, O. T., Rocha, V. G., Ferraro, C., Ni, N., D'Elia, E., Meille, S., Chevalier, J., Saunders, T., Peijs, T., Reece, M. J., Saiz, E. Using graphene networks to build bioinspired self-monitoring ceramics. *Nat. Commun.* **8**, 14425 (2017).

Supplementary Fig. 12 Surface chemistry of SBS/C (E/W=2:1), SBS/C (E) and SBS (W). XPS analysis of Sb 3d and C 1s peaks. Sb 3d peaks of (a) SBS/C (E/W = 2:1), (b) SBS/C (E), and (c) SBS/C (W); XPS analysis of C 1s peaks of (d) SBS/C (W).

Fig. 3 (c) XPS analysis of C 1s peaks of SBS/C (E/W = 2:1) and SBS/C (E).

Q3. The cyclic voltammograms were assigned based on in-situ XRD analysis and batteries research on SBS anode in lithium and sodium ion batteries (page 16). However, the presented references 12 and 20 correspond to the Sb_2S_3 studies as sodium ion batteries anodes. I would recommend to cite additionally the article of P.V. Prihodchenko et al. (Chemistry of Materials, 2012, 24, pp 4750–4757) describing the Sb_2S_3 @Graphene for a superior lithium battery anode.

A: Thanks for your carefully checking and recommendation. I have added the corresponding reference as you suggested. Please see the revision in manuscript blow:

“In the first cycle of few-layered SBS (NS-3-W), three cathodic peaks located at 0.78, 0.45 and 0.31 are assigned to the intercalation process and formation of solid-electrolyte interphase (SEI), the conversion reaction with sulphur atoms in SBS, and the alloying reaction of K with Sb, respectively, which are assigned based on in-situ XRD analysis and research on SBS anode in lithium and sodium ion batteries^{12,20,30}.”

12 Yu, D. Y. W., Prihodchenko, P. V., Mason, C. W., Batabyal, S. K., Gun, J., Sladkevich, S., Medvedev, A. G., Lev, O. High-capacity antimony sulphide nanoparticle-decorated graphene composite as anode for sodium-ion batteries. *Nat. Commun.* **4**, 2922 (2013).

20 Yao, S., Cui, J., Lu, Z., Xu, Z. L., Qin, L., Huang, J., Sadighi, Z., Ciucci, F., Kim, J. K. Unveiling the unique phase transformation behavior and sodiation kinetics of 1D van der Waals Sb_2S_3 anodes for sodium ion batteries. *Adv. Energy Mater.* **7**, 1602149 (2017).

30 Prikhodchenko, P. V., Gun, J., Sladkevich, S., Mikhaylov, A. A., Lev, O. L., Tay, Y. Y., Batabyal, S. K., Yu, D. Y. W., Conversion of hydroperoxoantimonate coated graphenes to Sb_2S_3 @graphene for a superior lithium battery anode. *Chem. Mater.* **24**, 4750–4757 (2012).

Q4. The elemental sulfur and SO_2 content in the SBS/C composite are discussed on the page 17 but all values are presented only in ESI. Some conclusions like " it should be noted that the element sulfur was detected in both electrodes and it shows much higher intensity/content in SBS/C than bulk SBS" should be supported with values of the sulfur content directly in the manuscript.

A: Thanks for your comments. As suggested, we tried to calculate the content of sulphur element based on the XPS technique. Because the normal XPS only can be used to analyse the surface chemistry of a material, however, the calculated values only could be used as a reference and comparison. Quantitative sulphur analysis by XPS was performed using the relative sensitivity factor method¹. The atomic concentration ratio of S^0 to S^{2-} could be obtained according to Equation (1) as shown below:

$$n_{\text{S}^0}/n_{\text{S}^{2-}} = (I_{\text{S}^0}/S_{\text{S}^0}) / (I_{\text{S}^{2-}}/S_{\text{S}^{2-}}) \quad (1)$$

where n is the number of atoms per cm^3 , I is the peak area, and S the sensitivity factor.

After fitting the S_{2p} peaks for SBS/C and SBS electrodes, the ratio of elemental S^0 to S^{2-} on the electrodes surface after the 50th charge can be calculated as 16.5% and 9.04% for SBS/C (E) and SBS (bulk), respectively (as shown in Supplementary Table 1 below).

1. Moulder, J. F., Sticle, W. F., Sobol, P.E., Bomben, K. D. Handbook of X-ray Photoelectron spectroscopy. Perkin-Elmer Co., Eden Prairie, MN (1992)

The revisions and discussions have been added to both the manuscript and the Supporting Information as follows:

“In addition, it should be noted that the element sulphur was detected in both electrodes, with its peak located at around 164 and 165 eV³², and the ratio of elemental S^0 to S^{2-} on the electrode surface after the 50th charge are 16.5% and 9.04 % for SBS/C (E) and SBS (bulk), respectively (Supplementary Table 1).” (Page 12, line 20)

Supplementary Table 1. XPS analysis of electrodes surface after 50th charge

Samples	Content after fitting				S (S ⁰)/S (S ²⁻)
	S (total)	S (-SO ₂ -)	S (S ²⁻)	S (S ⁰)	
SBS/C (E)	1	33%	57.76%	9.24%	16.5%
SBS (bulk)	1	45%	50.44%	4.56%	9.04 %

The quantitative sulphur analysis by XPS was performed using the relative sensitivity factor method¹. The atomic concentration ratio of S⁰ and S²⁻ could be obtained according to the equation (1) as shown below:

$$n_{S^0}/n_{S^{2-}} = (I_{S^0}/S_{S^0}) / (I_{S^{2-}}/S_{S^{2-}}) \quad (1)$$

where n is the number of atoms per cm³, I is the peak area, and S the sensitivity factor.

After fitting the S_{2p} peaks for SBS/C and SBS electrodes, each state of sulphur content could be estimated and the ratio of element S⁰ to S²⁻ on electrodes surface after 50th charge could be calculated as 16.5% and 9.04% for SBS/C (E) and SBS (bulk), respectively.

1. Moulder, J. F., Sticle, W. F., Sobol, P.E., Bomben, K. D. Handbook of X-ray Photoelectron spectroscopy. Perkin-Elmer Co., Eden Prairie, MN (1992)

Q5. The similar comment could be made regarding the discussion on the page 19 on the carbon content in the SBS/C composites. Values of the carbon content should be introduced into the text of the manuscript.

A: Thanks for your advice. The carbon content in each composite has been carefully studied via EDS and TGA. In the case of EDS, in order to give a reasonable results, we pressed the composite samples on the copper film, and collected more than 10 selected areas in low magnification in order to get the average values of carbon content. The carbon content in each composite is shown in Supplementary Fig. 19 after EDS analysis, with 3.82% in SBS/C (E:W = 1:1), 3.93% in SBS/C (E:W = 2:1), 3.62% in SBS/C (E:W = 6:1), 3.72% in SBS/C (E:W = 8:1), 4.17% in SBS/C (E:W = 16:1), and 4.94% in SBS/C (E:W = 32:1), indicating that there is not much differences in carbon content among these samples. At the same time, considering that Sb₂S₃ is converted into Sb₂O₄ when heated to 800 °C in air, the carbon contents in the composites can be calculated based on TGA data, and the calculated data are almost in agreement with the EDS results.

Supplementary Fig. 19 TG curves of the composite in air with a heating rate of $4\text{ }^{\circ}\text{C min}^{-1}$.

As suggested, the values of carbon content have been provided in the revised manuscript as follows:

“The carbon content in each composite has been investigated in Supplementary Fig. 19, (with 3.82% in SBS/C (E:W=1:1), 3.93% in SBS/C (E:W=2:1), 3.62% in SBS/C (E:W=6:1), 3.72% in SBS/C (E:W=8:1), 4.17% in SBS/C (E:W=16:1), and 4.94% in SBS/C (E:W=32:1)), indicating that there is not much difference in carbon content among these samples.” (Page 14, line 5)

Q6. I would suggest to revise the structure of the manuscript. The discussion part of the manuscript is in fact conclusions, but the chapter "results" actually contains "results and discussion".

A: Thanks for your suggestion. We prepared the manuscript based on the “Guide for Authors”, we also checked the previously published papers in *Nature Communications*, and found that the current structure is correct.

Reviewer #1 (Remarks to the Author):

The authors took a comprehensive consideration of the questions that were asked by the referee and gave corresponding reasonable explanations point by point. Additional information including related data and references has also been added. I think this manuscript is suitable for publishing in Nature communications.

Reviewer #2 (Remarks to the Author):

The authors have properly addressed the comments from the previous reviewers, and this manuscript is good for publication now.

Reviewer #3 (Remarks to the Author):

The revised manuscript can be published in Nature Communications as is.

Point by Point Responses:

Reviewer #1 (Remarks to the Author):

The authors took a comprehensive consideration of the questions that were asked by the referee and gave corresponding reasonable explanations point by point. Additional information including related data and references has also been added. I think this manuscript is suitable for publishing in Nature communications.

Response: We highly appreciate the reviewer's positive comments.

Reviewer #2 (Remarks to the Author):

The authors have properly addressed the comments from the previous reviewers, and this manuscript is good for publication now.

Response: We highly appreciate the reviewer's positive comments.

Reviewer #3 (Remarks to the Author):

The revised manuscript can be published in Nature Communications as is.

Response: We highly appreciate the reviewer's positive comments.